# Learning Compact Representations of Neural Networks using DiscriminAtive Masking (DAM)

**Jie Bu**[*]
Virginia Tech
jayroxis@vt.edu

**Arka Daw**[*]
Virginia Tech
darka@vt.edu

**M. Maruf**[*]
Virginia Tech
marufm@vt.edu

**Anuj Karpatne**
Virginia Tech
karpatne@vt.edu

## Abstract

A central goal in deep learning is to learn compact representations of features at every layer of a neural network, which is useful for both unsupervised representation learning and structured network pruning. While there is a growing body of work in structured pruning, current state-of-the-art methods suffer from two key limitations: (i) instability during training, and (ii) need for an additional step of fine-tuning, which is resource-intensive. At the core of these limitations is the lack of a systematic approach that jointly prunes and refines weights during training in a single stage, and does not require any fine-tuning upon convergence to achieve state-of-the-art performance. We present a novel single-stage structured pruning method termed DiscriminAtive Masking (DAM). The key intuition behind DAM is to discriminatively prefer some of the neurons to be refined during the training process, while gradually masking out other neurons. We show that our proposed DAM approach has remarkably good performance over a diverse range of applications in representation learning and structured pruning, including dimensionality reduction, recommendation system, graph representation learning, and structured pruning for image classification. We also theoretically show that the learning objective of DAM is directly related to minimizing the $L_0$ norm of the masking layer.

All of our codes and datasets are available `https://github.com/jayroxis/dam-pytorch`.

## 1  Introduction

A central goal in deep learning is to learn *compact* (or sparse) representations of features at every layer of a neural network that are useful in a variety of machine learning tasks. For example, in unsupervised *representation learning* problems [3], there is a long-standing goal to learn low-dimensional embeddings of input features that are capable of reconstructing the original data [44, 35, 24, 33]. Similarly, in supervised learning problems, there is a growing body of work in the area of *network pruning* [4], where the goal is to reduce the size of modern-day neural networks (that are known to be heavily over-parameterized [7, 51, 22, 45]) so that they can be deployed in resource-constrained environments (e.g., over mobile devices) without compromising on their accuracy. From this unified view of representation learning and network pruning, the generic problem of "learning compact representations" has applications in several machine learning use-cases such as dimensionality reduction, graph representation learning, matrix factorization, and image classification.

---

[*]These authors contributed equally to this work.

35th Conference on Neural Information Processing Systems (NeurIPS 2021).

A theoretically appealing approach for learning compact representations is to introduce regularization penalties in the learning objective of deep learning that enforce $L_0$ sparsity of the network parameters, $\theta$. However, directly minimizing the $L_0$ norm requires performing a combinatorial search over all possible subsets of weights in $\theta$, which is computationally intractable. In practice, a common approach for enforcing sparsity is to use a continuous approximation of the $L_0$ penalty in the learning objective, e.g., $L_1$-based regularization (or Lasso [42]) and its variants [39, 47]. While such techniques are capable of pruning individual weights and thus reducing storage requirements, they do not offer any direct gains in inference speed since the number of features generated at the hidden layers can still be large even though the network connectivity is sparse [4]. Instead, we are interested in the area of *structured network pruning* for learning compact representations, where the sparsity is induced at the level of neurons by pruning features (or channels) instead of individual weights.

While there is a growing body of work in structured network pruning [26, 28, 43, 14], the basic structure of most state-of-the-art (SOTA) methods in this area (e.g., ChipNet [43] and NetSlim [28]) can be described as training a learnable vector of mask parameters, $\mathbf{g} \in \mathbb{R}^n$, $\mathbf{g} = [g_1, g_2, ..., g_n]^\mathsf{T}$, which when multiplied with the features extracted at a hidden layer, $\mathbf{h} \in \mathbb{R}^n$, results in the pruned outputs of this layer, $\mathbf{o} = \mathbf{g} \odot \mathbf{h}$. Sparsity in the mask parameters is generally enforced using different approximations of the $L_0$ norm of $\mathbf{g}$ (e.g., use of Lasso in NetSlim and use of "crispness" loss in ChipNet). Despite recent progress in this area, current SOTA in structured pruning suffer from two key limitations. First, since most methods do not explicitly minimize the $L_0$ norm of $\mathbf{g}$ during pruning, they often suffer from *training instabilities*. In particular, most SOTA methods [28, 43] involve thresholding techniques to set small non-zero weights to zero leading to large drops in accuracy during the training process, as

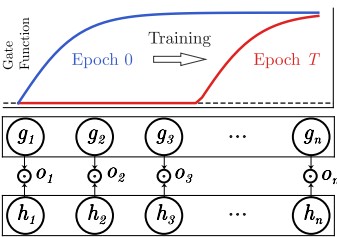

Figure 1: Illustration of Discriminative Masking. The gate function shifts to the "right" during training resulting in more zeros in the mask on convergence.

evidenced by our results in this paper. Second, once the pruning process is complete and we have converged at a compact network, most SOTA methods still need an additional step of *fine-tuning* the network in order to achieve reasonable accuracy. This is not only resource-intensive but there is also an on-going discussion on whether and how we should fine-tune [36] or rewind to initial weights [23] or train from scratch [29], making it difficult to prefer one approach over another.

At the core of these limitations is the lack of a systematic approach for structured network pruning that jointly prunes and refines weights during training in a single stage, and does not require any fine-tuning upon convergence to achieve SOTA performance. Notice that in existing methods for structured pruning, allowing every neuron $j$ to be pruned differently using an independent mask parameter $g_j$ only increases the number of learnable (or free) parameters in the learning objective, increasing the complexity of the problem. Instead, we ask the question: "Can we leverage the intrinsic symmetry of neural network initialization to design a pruning mask using the least number of free parameters?"

We present a simple solution to this question by proposing a new single-stage structured pruning method that learns compact representations while training and does not require fine-tuning, termed *DiscriminAtive Masking (DAM)*. The basic idea of DAM is to use a monotonically increasing gate function $\mathcal{G}$ for masking every neuron in a layer that only depends on the index $j = 1, 2, ..., n$ (or position) of the neuron in the layer and a scalar parameter $\beta$ to be learned during training (see Figure 1). At the start of training, the gate function admits non-zero values for all neurons in the layer, allowing all neurons to be unmasked (or active). As the training progresses, the gate function gradually shifts from "left" to "right" as a result of updating $\beta$ such that upon convergence, only a subset of neurons (on the extreme right) are active while all others are masked out. The key intuition behind DAM is to *discriminatively* prefer some of the neurons (on the right) to be refined (or re-adapted) during the training process for capturing useful features, while gradually masking out (or pruning) neurons on the left. This preferential pruning of neurons using a very simple gate function helps in regulating the number of features transmitted to the next layer.[*]

**Contributions:** We show that our proposed DAM approach has remarkably good performance over various applications, including dimensionality reduction, recommendation system, graph represen-

---

[*]This is somewhat analogous to how a physical dam regulates the flow of water by shifting a movable gate.

tation learning, and structured pruning for image classification, achieving SOTA performance for structured pruning. This shows the versatility of DAM in learning compact representations on diverse problems and network choices, in contrast to baseline methods for structured pruning that are only developed and tested for the problem of image classification [28, 43]. Our approach is single-stage, does not require fine-tuning (and hence has lower training time), and does not suffer from training instabilities. We also theoretically show that the learning objective of DAM is directly related to minimizing the $L_0$ norm of the discriminative mask, providing a new differentiable approximation for enforcing $L_0$ sparsity. Our approach is also easy to implement and can be applied to any layer in a network architecture. The simplicity and effectiveness of DAM provides unique insights into the problem of learning compact representations and its relationship with preferential treatment of neurons for pruning and refining, opening novel avenues of systematic research in this rapidly growing area.

## 2 Related Work

**Learning with Sparsity:** There is a long history of methods for enforcing compactness (or sparsity) in learned parameters [33, 42], where it is understood that compact hypotheses can not only result in more efficient data representations but also promote better generalizability [37]. This connection between compactness and generalization has also been recently explored in the context of deep learning [49, 1], motivating our problem of learning compact representations.

**Unstructured Pruning:** There are several methods that have been developed for unstructured network pruning [30, 10, 12, 38], where the goal is to prune individual weights of the network. A widely used technique in this field is referred to as the Lottery Ticket Hypothesis (LTH) [10], which suggests that a certain subset of weights in a network may be initialized in such a way that they are likely "winning tickets", and the goal of network pruning is then to simply uncover such winning tickets from the initialization of an over-parameterized network using magnitude-based pruning. Similar to our work, there exists a line of work in the area of *dynamic sparse training* for unstructured pruning [50, 13, 32, 9, 27] that gradually prunes the model to the target sparsity during training. However, while unstructured pruning methods can show very large gains in terms of pruning ratio, they are not directly useful for learning compact representations, since the pruned weights, even though sparse, may not be arranged in a fashion conducive to the goal of reducing the number of features.

**Structured Network Pruning:** The goal of structured pruning is to enforce sparsity over the network substructures (e.g., neurons or convolutional channels) that is directly relevant to the goal of learning compact representations. Most structured pruning SOTA methods need a three-stage process [16], i.e., training, pruning and finetuning, in order to get highly compressed models. Early exploration on the $L_1$ norm based filter pruning can be traced back to [26], followed by *Network Slimming* (NetSlim) [28], which uses a sparsity-inducing regularization based on $L_1$ norm. Following this line of work, a recent development in the area of network pruning is *ChipNet* [43], which introduces an additional *crispness loss* to the $L_1$ regularization, and achieves SOTA performance on benchmark tasks. However, network slimming can only work if the underlying architecture has batch normalization layers. It also has training instabilities due to soft-thresholding inherent to Lasso. ChipNet also iterates between soft- and hard-pruning, which is done after the training phase. Further, both these approaches require finetuning or retraining, which can be resource intensive.

Faster pruning schemes have also been explored by previous works. Bai et al [2] proposed a few-shot structured pruning algorithm utilizing filter cross distillation. The SNIP [25] took one step ahead by performing single-stage pruning. However, SNIP falls in the category of unstructured pruning. To the best of our knowledge, no existing structured pruning method has been demonstrated to achieve SOTA performance without finetuning. For example, while the method proposed in [30] can be potentially used as a single-stage structured pruning method with the help of group sparsity, the empirical analysis of the paper only focused on unstructured pruning.

**Deep Representation Learning**: Leveraging the power of deep neural networks, high-dimensional data can be encoded into a low-dimensional representation [18]. Such process is known as deep representation learning. A good representation extracts useful information by capturing the underlying explanatory factors of the observed input. The existence of noises, spurious patterns and complicated correlations among features make it a challenging problem. Previous work [44] show that an

Autoencoder architecture can be used for denoising images. To disentangle correlated features, VAE [19] and $\beta$-VAE [17] encourages the posterior distribution over the generative factors $q(z|x)$ to be closer to the isotropic Gaussian $\mathcal{N}(0, I)$ that explicitly promotes disentanglement of the latent generative factors. Driven by the idea of the autoencoder, the deep representation learning has been found remarkably successful in various of applications, e.g., graph representation learning [21, 6, 8, 31, 46, 34] and recommendation system [48].

However, most existing works in representation learning treat the embedding dimension as a hyperparameter, which can be crucial and difficult to choose properly. Small embedding dimensions cannot sufficiently represent the important information in the data that leads to bad representations. On the other hand, large embedding dimensions allow some level of redundancy in the learned representations, even picking up spurious patterns that can degrade model performances.

## 3   Proposed Approach

**Problem Statement:** Let us represent the general architecture of a neural network with $l$ layers as $\mathcal{F}(\boldsymbol{x}) = \Psi_l \circ \Psi_{l-1} \circ ... \Psi_2 \circ \Psi_1(\boldsymbol{x})$, where $\boldsymbol{x}$ are the input features to the network, $\Psi_i$ is the learnable function at layer $i$ that generates hidden features $\boldsymbol{h}_i = \Psi_i(\boldsymbol{h}_{i-1})$, and $f \circ g$ represents the composite function $f(g(\cdot))$. Let us also consider the "masked" version of this network where the features learned at every hidden layer, $\boldsymbol{h}_i$ are multiplied with a vector of mask values $\boldsymbol{g}_i$ using a mask function $M_i(\boldsymbol{h}_i) = \boldsymbol{g}_i \odot \boldsymbol{h}_i$. The masked neural network can then be described as $\mathcal{F}_{masked}(\boldsymbol{x}) = \Psi_l \circ M_{l-1} \circ \Psi_{l-1} \circ ... M_2 \circ \Psi_2 \circ M_1 \circ \Psi_1(\boldsymbol{x})$. The goal of learning compact representations can be framed as minimizing the following learning objective:

$$\min_{\theta} \mathbb{E}_{\boldsymbol{x} \sim \mathbb{D}_{tr}} \left[ \mathcal{L}\left( \mathcal{F}_{masked}(\boldsymbol{x}) \right) + \frac{\lambda}{l-1} \sum_{i=1}^{l-1} \|\boldsymbol{g}_i\|_0 \right], \tag{1}$$

where $\theta$ are the learnable parameters of the network, $\mathbb{D}_{tr}$ is the training set, $\mathcal{L}(.)$ is the loss function, and $\lambda$ is the trade-off parameter for enforcing $L_0$ sparsity in the mask values $\boldsymbol{g}_i$. Note that this formulation provides a unified view of both representation learning and structured network pruning, which differ in the choice of loss function (typically, reconstruction error for representation learning and cross-entropy for structured pruning), and the number of layers that are pruned (in representation learning, we only aim to prune the bottleneck layer while in structured pruning, we aim to prune all of the layers in the network, see Figure 2).

**DiscriminAtive Masking (DAM)**: Our proposed approach presents a novel perspective of structured pruning by introducing a new strategy of "ordering-and-masking." To understand the basic idea of this strategy, let us assume that we are given the ideal "ordering" of neurons in a layer that reflect their importance (i.e., neurons with higher order number are more useful for the learning task). Given this ordering, we can perform structured pruning by incrementally "masking" out neurons with lower order (i.e., setting their mask values to 0) before going to neurons with higher order during the training process. In other words, we *discriminatively* (or preferentially) focus on pruning neurons lower in the order, while retaining and refining the features learned at neurons higher in the order. This approach for masking can be realized using a monotonic gate function $\mathcal{G}$ that maps neuron order to mask values, i.e., lower order neurons are assigned smaller mask values than higher order neurons. By dynamically regulating the gate function during the course of training, we can control the percentage of neurons pruned at every training epoch.

Specifically, let us assign every neuron $j$ at layer $i$ with an "order number," $\mu_{ij}$. Before we describe how we arrive at these neuron orders, we first present the formulation of our proposed gate function $\mathcal{G}$, whose value at neuron $j$ monotonically increases with the order number of the neuron, $\mu_{ij}$, as follows:

$$g_{ij} = \mathcal{G}(\mu_{ij}, \alpha_i, \beta_i) = \text{ReLU}[\tanh\left(\alpha_i(\mu_{ij} + \beta_i)\right)], \tag{2}$$

where $\alpha_i$ is a constant scalar parameter (termed as the *steepness* parameter) that is not optimized during training, while $\beta_i$ is a 'learnable' scalar parameter (termed the *offset* parameter) that is the *only* parameter optimized during training to control (or regulate) the process of masking in DAM. Figures 2(c) and (d) show examples of gate function values (in red) at two different values of $\beta_i$. Note that using a single learnable parameter $\beta_i$ to implement the masking process introduces significant simplicity in the design of the learnable mask, in contrast to SOTA methods in structured

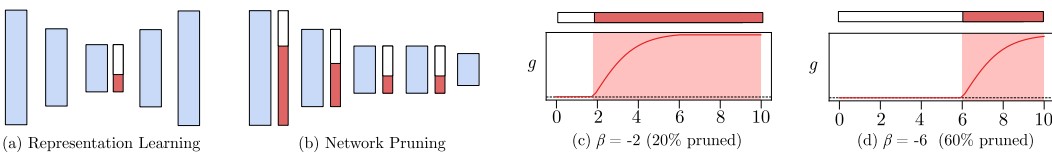

(a) Representation Learning    (b) Network Pruning    (c) $\beta = $ -2 (20% pruned)    (d) $\beta = $ -6 (60% pruned)

Figure 2: Illustration of the problems of representation learning (a) and structured network pruning (b). Blue blocks are layers in neural networks while red blocks show non-zero values in the mask layers. Figure (c) and (d) show two states of the gate function for $k = 10$. As an example, $\beta$=-2 in (c) prunes 20% of the neurons while $\beta$=-6 in (d) prunes 60% of the neurons.

pruning [28, 43] where a different masking parameter is trained for every neuron in the network. We hypothesize this simplicity of the gate function, in conjunction with the preferential pruning property of DAM, to result in remarkable gains in learning compact representations, as evidenced later in our experiments. Further, note that while we can use other monotonically increasing function to implement our gates, we found ReLU-tanh to perform best in practice.

**Assigning Neuron Orders** $\mu_{ij}$: We now discuss our approach for assigning neuron order numbers $\mu_{ij}$. Note that the order numbers only have to be assigned during the initialization step, and do not change during the training process. Further, note that every neuron is initialized with weights that are independent and identically distributed (because of the i.i.d nature of commonly used initialization methods in deep learning [41]). Hence, all neurons are symmetrically invariant to each other at initialization and any neuron has an equally good chance of refining themselves to capture useful features later in the training process as any other neuron. Hence, any random ordering of the neurons in the DAM approach would result in a similar pruning process (we empirically demonstrate the permutation invariance of $\mu_{ij}$ in Section 5). For simplicity, we consider a trivial choice of $\mu_{ij} = kj/n_i$, where $k$ is a constant parameter that determines the domain size of $\mu_{ij}$, and $n_i$ is the total number of neurons at layer $i$.

**How Does Reducing $\beta_i$ Enforce $L_0$ Sparsity?** Let us first understand how mask values $\boldsymbol{g}_i$ respond to the only learnable parameter $\beta_i$. Figure 2 (c) shows an example of $g_i$ when $\beta_i = -2$. At this stage, we can see that only a small number of neurons have zero gate values (white). As we reduce $\beta_i$ to $-6$ in Figure 2 (d), we can see that the gate moves towards the 'right', resulting in more excessive pruning of neurons. In general, the number of non-zero values of the gate function (and hence, its $L_0$ norm) is directly related to $\beta_i$ as follows:

$$\|\boldsymbol{g}_i\|_0 = \sum_{j=1}^{n_i} \mathbb{1}(g_{ij} > 0) = \sum_{j=1}^{n_i} (1 - \mathbb{1}(\mu_{ij} \leq \beta_i)) = \lceil n_i(1 + \beta_i/k) \rceil, \quad \text{for} \quad \beta_i \geq -k, \quad (3)$$

where $\mathbb{1}$ indicates the identity operator and $\lceil \cdot \rceil$ is the ceiling operator. In order to back-propagate gradients, we use a continuous approximation of Equation (3) (by dropping the ceiling operator) for regularization as $\|\boldsymbol{g}_i\|_0 \approx n_i(1 + \beta_i/k)$. Hence, minimizing $L_0$ norm of $\boldsymbol{g}_i$ is directly proportional to minimizing $\beta_i$ with a scaling factor of $n_i/k$.

**Learning Objective of DAM:** While minimizing $\beta_i$ with scaling factors proportional to $n_i$ can explicitly minimize $L_0$ norm of the gate values, we found in practice that layers with smaller number of neurons $n_i$ still require adequate pruning of features for learning compact representations proportional to layers with large $n_i$. Hence, in our learning objective, we drop the scaling factor and directly minimize the sum of $\beta_i$ across all layers in the following objective function.

$$\min_{\theta, \boldsymbol{\beta}} \mathbb{E}_{x \sim \mathbb{D}_{tr}} \left[ \mathcal{L}\left(\mathcal{F}_{masked}(\boldsymbol{x})\right) + \frac{\lambda}{l-1} \sum_{i=1}^{l-1} \beta_i \right], \quad (4)$$

where, $\boldsymbol{\beta}$ is the set of all $\beta_i$'s in the network, $\{\beta_1, \beta_2, \ldots, \beta_{l-1}\}$.

**Theoretical Analysis of DAM:** Analysis of the dynamics of the gate functions and additional specifications for choosing $\alpha_i$ and the gate function are provided in Appendix A.

**Parameter Specification and Implementation Details:** In all of our experiments, we used $\alpha_i = 1$ and $k = 5$. In our structured network pruning experiments, we used a cold-start of 20 epochs (i.e., the $\beta_i$'s were frozen for the duration of cold-start), so as to allow the leftmost neurons to undergo

some epochs of refinement before beginning the pruning process. We also set the initial value of $\beta_i$ to 1, which can also be thought of as another form of cold-starting (since pruning of a layer only starts when $\beta_i$ becomes less than zero).

## 4 Experiments on Representation Learning Problems

### 4.1 Results on Dimensionality Reduction (DR) Problems

**Problem Setup:** We evaluate the effectiveness of DAM in recovering the embeddings of synthetically generated dimensionality reduction problems. The general form of the problem is expressed by (5). Suppose $\Omega \in \mathbb{R}^r$ and all of its $r$ dimensions are independent from each other. A $d$-dimensional data $X$ can be expressed using a transformation $\Psi : \mathbb{R}^r \longrightarrow \mathbb{R}^d$ defined on Hilbert space, and $X = \Psi(\Omega)$, $d > r$. Let $M$ be the learnable mask, we formulated the dimensionality reduction problem as:

$$\min_{\Theta} \left\{ \|\mathcal{F}_D \circ M \circ \mathcal{F}_E(X) - X\|_F^2 + \lambda\beta \right\}, \quad \Theta = \{\theta_E, \theta_D, \beta\} \tag{5}$$

The $\mathcal{F}_E, M, \mathcal{F}_D$ are trained end-to-end using gradient descent. In our experiments, $\Omega$ is sampled from a isotropic normal distribution, i.e., $\Omega \sim \mathcal{N}_r(\mathbf{0}, I_r)$. We let $\Psi$ to use the same structure as the decoder $\mathcal{F}_D$ with randomly generated parameters $\theta_\Psi$, i.e., $\Psi = \mathcal{F}_D|_{\theta_D=\theta_\Psi}$, so that the minimum number of dimensions needed in the encoded representation for the decoder to reconstruct $X$ is $r$.

*Linear DR:* We test DAM for removing linear correlations in a given rank-deficient matrix. In this case, $\Psi$ is a matrix representing a linear projection from $\mathbb{R}^r$ to $\mathbb{R}^d$. We use two real matrices as the encoder and decoder and train the DAM to find the full-rank representation of $X$.

*Nonlinear DR:* To test the ability of DAM for disentangling nonlinearly correlated dimensions, we present two cases: (i) $\Psi$ is a polynomial kernel of degree 2 (we use a QRes layer from [5]); (ii) $\Psi$ is a nonlinear transformation expressed by a neural network. We use a deep neural network as the encoder with sufficient complexity.

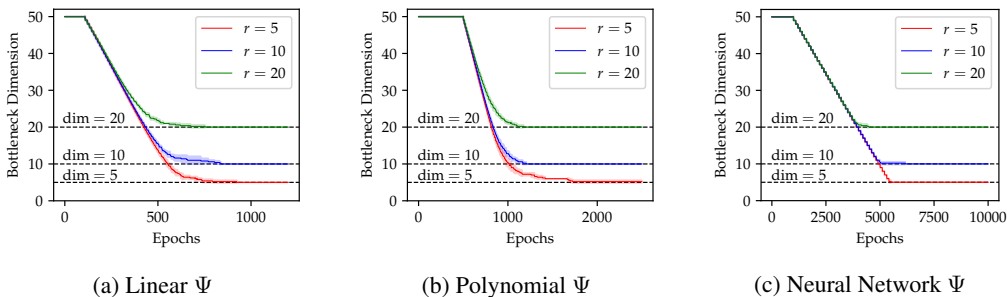

(a) Linear $\Psi$    (b) Polynomial $\Psi$    (c) Neural Network $\Psi$

Figure 3: Bottleneck dimension (of nonzero entries) vs number of epochs during training.

**Observation:** Figure 3 shows the convergence of bottleneck dimensions over the training epochs for different synthetic data with varying sizes of underlying factors ($r$). The curves and shades show the mean and standard deviation of five different runs. We can see that DAM consistently uncovers the exact dimension of the underlying factors $r$. We provides details of in-depth sensitivity analysis of DAM results based on its hyper-parameters (e.g., learning rate and $\lambda$) in Appendix C.

**Theoretical Analysis of DAM for linear DR:** We theoretically show that DAM converges to the optimal solution for the linear DR case (see Appendix B for details).

### 4.2 Recommendation System Results

**Problem Setup:** We consider the state-of-the-art method for recommendation system problems (IGMC [48]), which transforms the rating matrix completion task to a link prediction problem, where the ratings are interpreted as links between users and items.

In particular, IGMC generates 128-dimensional embeddings to represent the enclosing subgraph and further avoids overfitting using a dropout layer with a dropout rate of 0.5. We replaced the dropout

Table 1: Results of IGMC with and w.o. DAM on recommendation system tasks.

| | IGMC | | IGMC-DAM | |
| | RMSE | Dimension | RMSE | Dimension |
| --- | --- | --- | --- | --- |
| Flixter | $0.8715 \pm 0.0005$ | 128 | $\mathbf{0.8706 \pm 0.0003}$ | **32** |
| Douban | $0.7189 \pm 0.0002$ | 128 | $\mathbf{0.7183 \pm 0.0004}$ | **17** |
| Yahoo-Music | $19.2488 \pm 0.0123$ | 128 | $\mathbf{19.0166 \pm 0.0444}$ | **83** |

layer with a DAM layer in the IGMC to reduce the dimension of the learned representations. We train our IGMC-DAM model for 100 epochs under the same training configurations.

**Observation:** Table 1 shows that DAM successfully reduces the dimensions of the learned representations without any increase in errors, demonstrating that DAM is able to boost the performance of IGMC by learning compact representations over the target graph.

### 4.3 Graph Representation Learning Results

**Problem Setup:** Following the previous experiment, we further explore the effectiveness of DAM in learning compact graph representations. Our goal is to learn low-dimensional embeddings for each node in a graph that captures the structure of interaction among nodes. A simple graph autoencoder, e.g., GAE [21], uses a graph convolutional network (GCN) [20] as an encoder and an inner product as a decoder. The encoder calculates the embedding matrix $Z$ from the node feature matrix $X$ with the adjacency matrix $A$, and the decoder reconstructs a adjacency matrix $\hat{A}$ such that: $\hat{A} = \sigma(ZZ^{\top})$, with $Z = \text{GCN}(X, A)$. The reconstruction loss, $\text{BCE}(A, \hat{A})$ is backpropagated to train the GAE model. To reduce the dimension of the learned representation, we add a DAM layer in a GAE after the encoder (GAE-DAM).

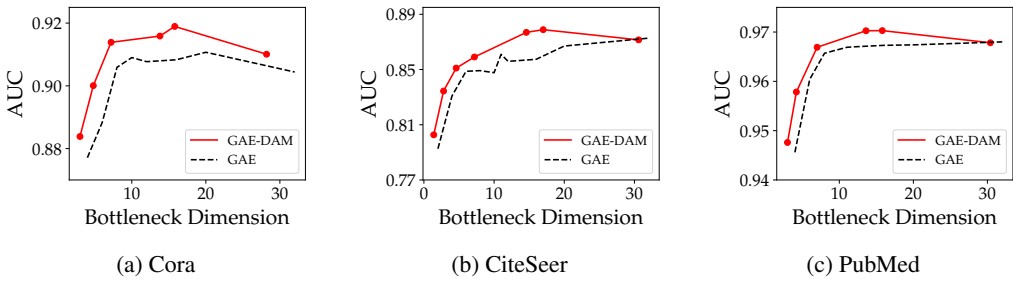

| (a) Cora | (b) CiteSeer | (c) PubMed |

Figure 4: Link prediction performance for Cora, CiteSeer and PubMed Dataset. Competing models are GAE, GAE-DAM.

**Observation:** Figure 4 shows that DAM based GAE method is able to learn meaningful structural information in compact latent embeddings. DAM improves the link prediction performance of the simple GAE for Cora, CiteSeer and PubMed dataset by learning compact representations.

### 4.4 Representation Learning Results for MNIST Dataset

**Problem Setup:** We further evaluate DAM on the problem of dimensionality reduction using simple auto-encoders on the MNIST dataset, and also compare the effectiveness of DAM (that directly enforces $L_0$ sparsity) with an ablation of our approach that instead minimizes the $L_1$ norm (similar to Lasso). We vary the trade-off parameter $\lambda$ to compare the sparsity of the learned representations between DAM and $L_1$-norm based ablation method.

**Observation:** Figure 5 shows that DAM is able to achieve lower reconstruction error as well as higher $F1$ scores over the same bottleneck dimensions than $L_1$ based method. In Figure 5 (c), we can see that DAM shows a near-linear descending trend of bottleneck dimensions as $\lambda$ increases (further evidence presented in Appendix E1). In contrast, the $L_1$ based method has a saturating effect at large $\lambda$ values since the weights come close to zero but require some thresholding to be pruned. This shows that DAM is amenable to learning highly compact representations in contrast to $L_1$ based methods.

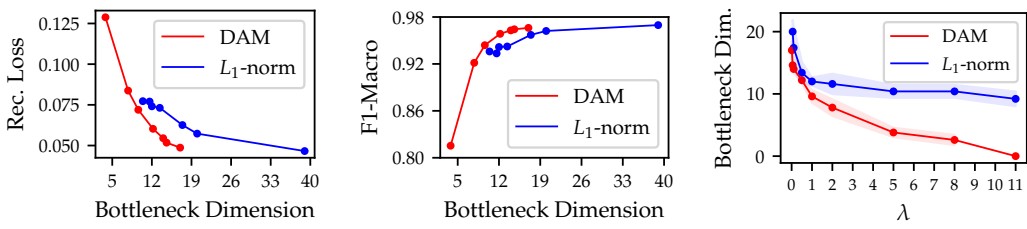

Figure 5: Performance comparison plots of DAM with $L_1$-norm method on MNIST dataset.

# 5 Experiments on Structured Network Pruning

**Evaluation Setup** Here we demonstrate the effectiveness of DAM on the task of structured network pruning. We compared the performance of our proposed DAM with two SOTA structured pruning algorithms, Network Slimming (Net-Slim) [28] and ChipNet [43]. Net-Slim is well-established SOTA that is widely used as a strong baseline, while ChipNet is the latest SOTA representing a very recent development in the field. ChipNet and Net-Slim are both pretrained for 200 epochs, while the DAM performs single-stage training and pruning in the same number of epochs. Since, our DAM approach does not require additional fine-tuning, we imposed a limited budget of 50 epochs for fine-tuning the SOTA methods. Note that ChipNet also has a pruning stage which involves 20 additional epochs. We evaluate the performance of these methods on the PreResnet-164 architecture on benchmark computer vision datasets, CIFAR-10 and CIFAR-100. Additional evaluation setup details are in Appendix E2.

**Performance and Running time comparison**: Figures 6a and 6b demonstrate the performance of different network pruning methods for various pruning ratios. We observe that for both datasets, DAM is able to outperform Net-Slim by a significant margin specially when the models are sparse. Additionally, DAM is able to achieve similar or slightly better performance than ChipNet with no additional fine-tuning. We also compare the total running time for pruning across the different models (divided into the three categories) in Figure 6c. We can see that the training time of the three methods are almost the same, with ChipNet being slightly small. However, the running time for the pruning and fine-tuning stages for DAM are both zero. Net-Slim also does not involve additional pruning after training. However, ChipNet involves 20 epochs of pruning which is significant and is almost comparable to its pretraining time. Finally, comparing the total running time taken by each of the structured pruning methods, we observe that DAM is significantly faster than the current SOTA counterparts owing to its single-stage pruning approach.

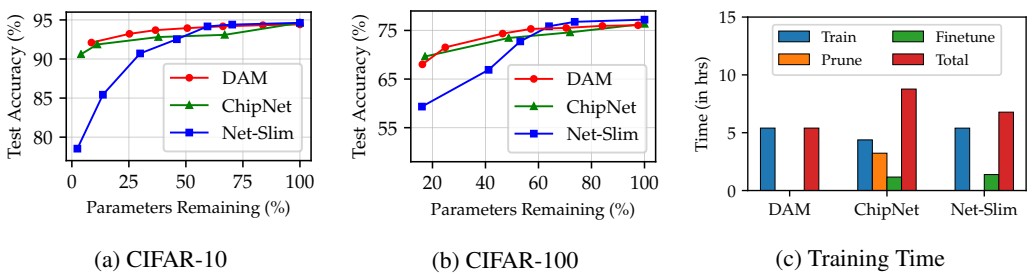

Figure 6: Performance and running time comparison of DAM with state-of-the-art structured pruning methods on PreResNet-164 for various parameter pruning ratios.

**Stability Analysis**: We further analyse the stability of our proposed DAM approach for network pruning through visualization of the training dynamics. We observe that the training cross-entropy (CE) loss and the validation CE loss are very similar to what we expect from a Preresnet model trained on benchmark vision datasets with learning rate changes at 100 and 150 epochs, respectively. We further notice that the total training loss for the DAM is also very stable and does not show any sudden variations, leading to steady convergence. Finally, from the convergence plot for $\mathbb{E}[\beta_i]$ (green-dashed line), we can see that our pruning does not involve any sudden steps and happens in a continuous and steady rate throughout the training stage.

The training dynamics of the ChipNet (blue shaded region in the middle panel) is exactly the same as training vanilla vision models since it does not involve sparsity constraints. However, as we move to the pruning stage, we notice a sharp rise in both training and validation losses (white region). This is due to the use of a different optimizer AdamW as opposed to SGD which was used in the training stage. Further, a very interesting phenomenon happens in the fine-tuning stage (red shaded region) where the training and validation losses increases for the first 25 epochs and then takes a sharp decrease once the learning rate is adjusted. This suggests that after the pruning stage, finetuning the model with large learning rate forces the model to make large updates in the initial epochs (akin to training from scratch), which further drives the model out of the local minima. Once the learning rate is adjusted, the model is able to slowly progress towards a new local optima and ultimately converges to the fine-tuned minima.

Finally, both the training and fine-tuning dynamics of the Net-Slim (rightmost panel) appears to be highly unstable. This suggests that the pre-training with sparsity loss is not stable and can lead to large fluctuations in the training loss. Also, note that Net-Slim does not have a validation set as it instead uses the test set to choose the best model. We have refrained from using the best model by looking at the performance on the test set, and instead evaluate the model from the last epoch.

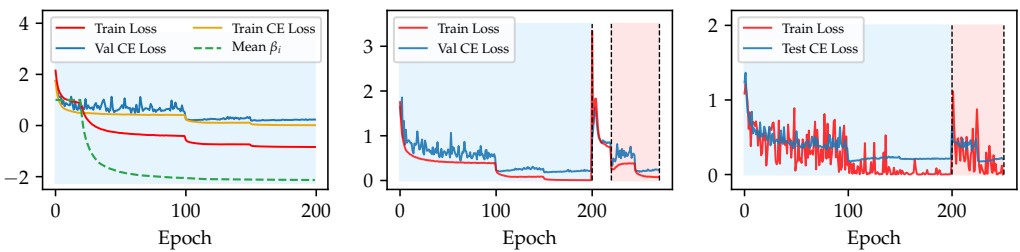

Figure 7: Stability analysis for DAM (left), ChipNet (middle) and Network Slimming (right) through visualization of the loss curves. Blue, white and red shades denote the training, pruning and fine-tuning stages respectively.

**Permutation Invariance**: To test the effect of permuting the neuron order on the performance of DAM, Table 2 provides results across five randomly permuted values of $\mu_{ij}$ across every neuron in the network, while using the same initialization as previous experiments. We can see that DAM is invariant to permutations of neuron order during initialization, validating our simple choice of assigning neuron ordering based on their index values.

Table 2: Permutation Invariance results on PreResNet-164 on CIFAR-10. RSD denotes the relative standard deviation.

|  | Run 1 | Run 2 | Run 3 | Run 4 | Run 5 | RSD |
|---|---|---|---|---|---|---|
| Test Accuracy | 94.11% | 93.86% | 93.71% | 93.93% | 93.52% | 0.0024 |
| Channels Pruned | 43.31% | 43.14% | 43.31% | 43.14% | 43.02% | 0.0029 |
| Parameters Pruned | 64.01% | 63.40% | 63.97% | 63.64% | 63.37% | 0.0048 |

**Additional Results:** To illustrate generalizibility we provide additional network pruning results for more networks such as VGG-19, PreResnet-20, PreResNet-56 and PreResnet-110 on CIFAR datasets and LeNet-5 on MNIST in Appendix E3. To demonstrate decorrelated kernels obtained after pruning using DAMs we use CKA similarity [22]. The results can be found in Appendix E3. For extreme values of $\lambda$, we demonstrate that DAM is able to pruning entire bottleneck layers for ResNet architectures by visualizing the pruned architectures in Appendix E4.

## 6   Discussion, Open Questions, and Future Works

The effectiveness of DAM in learning compact representations provides a novel perspective into structured network pruning and raises several open questions about its relationships with other concepts in this area. In the following, we list some of these connections and describe the limitations and future extensions of our work.

**Connections with Lottery Ticket Hypothesis**: Our work builds on the intuition of discriminative masking, i.e., we preferentially favor some neurons to be refined while favoring some other neurons to be pruned. This has interesting connections with the lottery ticket hypothesis (LTH) [10], where some subsets of initialized weights in the network are hypothesized to contain "winning tickets" that are uncovered after a few epochs of training. While there is some open discussion on whether the winning tickets exist even before initialization [11, 15], there are several studies supporting this hypothesis [38, 12]. Our results attempts to throw light on the question: "can we find winning tickets if we discriminatively search for it in a certain subregion of the network?" Further, we are able to show that the results of DAM are invariant to random permutation of the neuron indices at initialization, since all neurons receive identically distributed weights. Along these lines, we can also explore if there are certain ordering of neurons (e.g., in accordance with their likelihood of containing winning tickets revealed through LTH) that can perform better than random ordering in DAM.

**Connections with Dropout:** Another interesting connection of DAM is with the popular regularization technique of Dropout [40] that randomly drops out some neurons during training to break "co-adaptation" patterns in the learned features so as to avoid overfitting. Essentially, by randomly dropping a neuron at some epoch of training, the other neurons are forced to pick up the learned features of the dropped neuron, thus resulting in the learning of robust features. This is similar to the "re-adaptation" of weights at the active neurons at some epoch of DAM training, while the mask value of the neurons in the transitioning zone (when $0 < g_j < 1$) are gradually dropped by the shifting of the gate function. This motivates us to postulate the *weight re-adaptation hypothesis* as a potential reason behind the effectiveness of DAM in learning compact representations, which requires further theoretical justifications.

**Budget-aware Variants of DAM**: One promising future extension of our current DAM formulation would be to make it budget-aware, i.e., to stop the training process once we arrive at a target level of sparsity. Note that there is a direct correspondence between the value of the learnable gate offset parameter $\beta_i$ (that is minimized at every epoch) and the resulting $L_0$ sparsity (see Equation 3). Hence, a simple budget-aware extension of our current DAM formulation would be to start from a large value of lambda (to provide sufficient scope for aggressive pruning) and keep monitoring the level of sparsity at every layer during the training process. As soon as the target sparsity mark is achieved, we can freeze $\beta_i$'s at every layer thus essentially halting the pruning process (note that the network architecture becomes immutable if $\beta_i$'s is fixed).

**DAM Variants for Pre-trained Networks:** One of the fundamental assumptions of our current DAM formulation is that all neurons are symmetrically invariant to each at initialization, and hence we can simply use random neuron ordering for discriminative masking. While this assumption is valid for "training from scratch", this may not hold for neurons in a pre-trained network. Future extensions of DAM can include advanced ways of ordering neurons that do not rely on the above assumption and hence can even be used with pre-trained networks. For example, we can order neurons based on the magnitudes of weights of every neuron or the value of mutual information between neuron activations and outputs. This would open novel avenues of research in structured network pruning by finding effective neuron orderings for different initializations of network weights.

# 7 Acknowledgement

This work was supported by the NSF Eager Grant #2026710.

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
