# Appendix:
# Learning Compact Representations of Neural Networks using DiscriminAtive Masking (DAM)

## A   Analysis of the DAM Gate Function Dynamics During Training

In this section, we theoretically analyze the dynamics of the DAM mask $g_i$ at the $i$-th layer as the training process unfolds. This can be characterized by the movement of the gate function (or gate) during the training process, which is solely dominated by changes in the learnable offset parameter $\beta_i$.

### A.1   Gradients of DAM Learning Objective w.r.t. $\beta_i$

Building on the notations introduced in Section 3 of the main paper, let us denote the mask value $g_{ij}$ at the $j$-th neuron at $i$-th layer as a function of $\beta_i$, i.e., $\xi_j : (\beta_i, j) \to g_{ij}$, where

$$g_{ij} = \xi_j(\beta_i, j) = \max\left[\tanh\left(\alpha_i\left(kj/n_i + \beta_i\right)\right), 0\right] \tag{1}$$

Let $f$ denote the continuous function expressed by our neural network with learnable parameters $\Theta$. The loss function for training the neural network for the target task can then be denoted as $L = \mathcal{L}(f(x, \Theta, \beta_i))$ (e.g., cross-entropy loss for supervised structured pruning problems and reconstruction error for representation learning problems), where $x$ denotes the input features to the neural network. Using gradient descent methods with a learning rate of $\eta$, the expected update formula of $\beta_i$ in DAM is given by:

$$\Delta\beta_i = -\eta\,\mathbb{E}_{x\sim\mathcal{D}_{tr}}\left[\nabla_{\beta_i}\mathcal{L}(f(x,\Theta,\beta_i)) + \lambda\nabla_{\beta_i}\beta_i/(l-1)\right] \tag{2}$$

$$= -\eta\,\mathbb{E}_{x\sim\mathcal{D}_{tr}}\left[\nabla_{\beta_i}\mathcal{L}(f(x,\Theta,\beta_i))\right] - \eta\lambda/(l-1) \tag{3}$$

Let $h_i$ be the layer output before applying the DAM mask, and the masked output be represented as $o_i = h_i \circ g_i$ after applying the gate. The gradient of the loss w.r.t. $\beta_i$ can be obtained by applying the chain rule of differentiation as follows:

$$\nabla_{\beta_i}\mathcal{L}(f(x,\Theta,\beta_i)) = \frac{\partial\mathcal{L}(f(x,\Theta,\beta_i))}{\partial o_i}\frac{\partial o_i}{\partial g_i}\frac{\partial g_i}{\partial\beta_i} = \sum_{j=1}^{n_i}\frac{\partial\mathcal{L}(f(x,\Theta,\beta_i))}{\partial o_{ij}}\frac{\partial o_{ij}}{\partial g_{ij}}\frac{\partial g_{ij}}{\partial\beta_i} \tag{4}$$

Let us analyze this gradient by observing the last term of this equation, $\partial g_{ij}/\partial\beta_i$. For the $j$-th neuron, $\partial g_{ij}/\partial\beta_i = 0$ if and only if $\partial\xi_j(\beta_i)/\partial\beta_i = 0$. Since $tanh(z)$ has non-zero gradients for $z > 0$, the gradient of $\xi_j(\beta_i)$ is 0 only when $kj/n_i + \beta_i \le 0$, i.e., the mask value of the neuron is 0 (or in other words, it is *deactivated* or dead). Let us denote the set of all neuron indices with non-zero mask values (also referred to as *active* neurons) as $\mathcal{J}$. Equation 4 can then be simplified as:

$$\nabla_{\beta_i}\mathcal{L}(f(x,\Theta,\beta_i)) = \alpha_i\sum_{j\in\mathcal{J}}\underbrace{\frac{\partial\mathcal{L}(f(x,\Theta,\beta_i))}{\partial o_{ij}}h_{ij}\left(1 - g_{ij}^2\right)}_{q_{ij}}, \tag{5}$$

$$= \alpha_i\sum_{j\in\mathcal{J}}q_{ij}, \tag{6}$$

where $q_{ij}$ represents the contribution of the $j$-th neuron to the gradient of the loss term with respect to $beta_i$. We can make the following two observations: (i) only those neurons that are active (i.e., have non-zero mask values) have a contribution towards updating $\beta_i$ and moving the gate function. (ii) If the mask value of a neuron is 1 (i.e., $g_{ij} = 1$), then their contribution to the gradient of the loss w.r.t. $\beta_i$ is again 0. It shows that the neurons that play an important role in the dynamics of the gates are the ones with non-zero activation $h_{ij}$ and mask values that have not saturated to 1 (i.e.,

$g_{ij} < 1$). We name these neurons as *support* neurons and their position in the ordering of neurons as the *transitioning zone* of the gate function. Similarly, neurons with zero mask values are termed as *deactivated* neurons and the neurons with mask values close to 1 as *privileged* neurons (since they are never turned off).

## A.2 Equilibrium of DAM Gate Function upon Convergence

We next study the properties of the equilibrium solution of $\beta_i$ that we arrive upon convergence of the DAM training process. Suppose that we have converged to $\beta_i^*$ for the offset parameter for the $i$-th layer. This would mean that the gradient of the DAM learning objective w.r.t. $\beta_i^*$ would be equal to 0 as follows:

$$\nabla_{\beta_i^*} \mathcal{L}(f(x, \Theta, \beta_i^*)) + \lambda/(l - 1) = 0 \tag{7}$$

Substituting the value of $\nabla_{\beta_i} \mathcal{L}(f(x, \Theta, \beta_i))$ from Equation (6) and rearranging terms, we get

$$\sum_{j \in \mathcal{J}} q_{ij} = -\lambda/(\alpha_i(l - 1)). \tag{8}$$

Since $\lambda, \alpha_i > 0$, the equilibrium exists only when the sum of $q_{ij}$ is negative. This happens when decreasing the masked outputs $o_{ij}$ of the support neurons in the transitioning zone leads to an increase in the loss function, signifying that any further pruning can lead to loss of accuracy. In other words, the training dynamics of the gate function stops when the features learned at the support neurons are useful enough that their pruning is detrimental to the generalization performance of the network.

## A.3 Additional Remarks on the Effects of $\alpha_i$ and $\lambda$ On DAM Convergence

Equation 8 also implies that a large value of $\alpha_i$ or a low value of $\lambda$ may make the equilibrium easy to reach, thus leading to *premature convergence* to a pruned network that has not been fully trained to capture refined features at its support neurons. This is supported by our empirical observations that large values of $\alpha_i$ tends to prevent DAM layers to reach higher sparsity. We thus set $\alpha_i = 1$ in all our experiments. On the other hand, the choice of $\lambda$ also plays a big role as we observed in our experiments (e.g., as is shown in Section C via hyperparameter sensitivity analysis).

Further note that the masking dynamics of DAM involve a smooth transition from an unpruned network to a pruned network where at every training epoch, some of the neurons (or channels) gradually die out while the rest remain unaffected. The neurons at the edge of being dropped out are gradually assigned low gate values such that the other neurons can slowly re-adapt themselves to pick up or recover the features that are being dropped out. However, steep gate function choices (that have narrow transitioning zones) may cause this transition to be too fast such that the network suffers a drop in accuracy as neurons on the edge are dropped while the other neurons do not have sufficient time to recover the dropped features. In our formulation, choosing a very large value of $\alpha_i$ can make the gate function too steep for effective pruning (we thus choose $\alpha_i = 1$). For other alternate choices of gate function, $\xi_j(\beta_i)$, than what we used in our current implementation, similar considerations need to be observed to avoid the gate function from becoming too steep.

# B Theoretical Results And Technical Proofs For Section 4

We empirically observed in Section 4 of the main paper that for the linear dimensionality reduction case, we always converged to the same solution for a constant setting of hyper-parameters regardless of the random initialization of the neural network. To theoretically understand if DAM is capable of converging to the optimal solution (where the pruned bottleneck dimension is exactly same as the rank of the input data matrix), we provide further theoretical analysis for the ability of DAM to perform linear dimensionality reduction (DR) in the following.

**Problem Statment**: Let $G = \mathrm{diag}(\boldsymbol{g})$, where $\boldsymbol{g} \in \mathbb{R}^n$ is the vector of mask values at the bottleneck layer given by

$$\boldsymbol{g} = \max\left[\tanh(\boldsymbol{\mu} + \beta \mathbf{1}), 0\right] \tag{9}$$

and the $j$-th element in $\boldsymbol{\mu}$ is given by $\mu_j = kj/n$, while $\beta \in \mathbb{R}$ is a scalar learnable parameter. Let the real matrices $A_1, A_2$ define two linear transformations $A_1 : \mathbb{R}^n \longrightarrow \mathbb{R}^d$ and $A_2 : \mathbb{R}^d \longrightarrow \mathbb{R}^n$. Let

us consider a bounded input data matrix, $X \in \mathbb{R}^{d \times N}$, that is rank-deficient (with rank $m$), where $N$ is the number of data samples and $d$ is the number of features, $N \gg d > m$. Then, the input $X$ and its reconstruction $\hat{X}$ are related by the following equation:

$$\hat{X} = A_1 G A_2 X. \tag{10}$$

Let $\Theta = \{A_1, A_2, \beta\}$ be the set of learnable parameters. For a given $X$, let us define the multi-objective minimization problem as

$$\min_{\Theta} \left( \left\| \hat{X} - X \right\|_F^2, \beta \right), \tag{11}$$

where $\|.\|_F^2$ denotes the Frobenius norm of a matrix. We are interested in reaching a Pareto optimal solution with a predefined trade-off hyper-parameter between the two objectives using the gradient descent algorithm.

**Optimization Scheme**: Let $L = \left\| \hat{X} - X \right\|_F^2 + \lambda \beta$ and initial values for the learnable parameters be $A_1^{(0)}, A_2^{(0)}, \beta^{(0)}$, where $\lambda > 0$ is the trade-off hyper-parameter. Using gradient descent (GD), for every parameter $\theta \in \Theta$, the updating rule for the $t$-th iteration can be written as

$$\theta^{(t+1)} = \theta^{(t)} - \eta (\boldsymbol{\nabla}_\theta L)^{(t)} \tag{12}$$

where $\eta \in \mathbb{R}^+$ is the step size. We assume that we allow the network to perform sufficient number of iterations of gradient updates before it reaches convergence.

**Theorem 1.** *(Existence of Optimal Solution) Assuming all the $m$ singular values of $X$ are non-trivial (i.e., they are all greater than some positive value $\epsilon$), there exists $\lambda > 0$ such that at the minima solution of $\mathcal{L}$, $\hat{X} = X$ and the number of non-zero entries of $\boldsymbol{g}$ equals $m$.*

*Proof:* Let $\tilde{A}_2 = G A_2$. Then, we can rewrite $\hat{X}$ as a function of $\tilde{A}_2$ as follows

$$\hat{X} = A_1 \tilde{A}_2 X. \tag{13}$$

Using this value of $\hat{X}$ as a function of $\tilde{A}_2$, we can decouple the combined learning objective into the following two optimization problems:

$$\left( \min_{\{A_1, \tilde{A}_2\}} \left\| A_1 \tilde{A}_2 X - X \right\|_F^2, \quad \min_{\beta} \lambda \beta \right), \tag{14}$$

where the first term only depends on the variables $A_1, \tilde{A}_2$, while the second term only depends on $\beta$. Optimizing the second term is trivial since it is a linear function of $\beta$. On the other hand, the first term has its minimum value at

$$A_1 \tilde{A}_2 X = X$$
$$A_1 = \tilde{A}_2^+ \tag{15}$$

where $\tilde{A}_2^+$ is the *Moorse-Penrose Inverse* of $\tilde{A}_2$. Moreover, whenever $A_1 = \tilde{A}_2^+$, the following relationship would also hold

$$\operatorname{rank}(\tilde{A}_2) \geq \operatorname{rank}(A_1 \tilde{A}_2 X) = \operatorname{rank}(X) = m. \tag{16}$$

and $\operatorname{rank}(\tilde{A}_2) = \operatorname{rank}(G A_2) \leq \operatorname{rank}(G)$. Thus, $\operatorname{rank}(G) \geq m$. Assuming the diagonal matrix $G$ has $m'$ nonzero entries, then $m' = \operatorname{rank}(G) \geq m$. This means that the lowest value that $m'$ can take is $m$, and we want to study if there exists some setting of $A_1, \tilde{A}_2$, and $\beta$ that leads to $m' = m$. We specifically explore the setting where the rank of $G$ (the number of non-zero values of $\boldsymbol{g}$) is equal to $m$. Note that in our DAM formulation, the number of non-zero entries of the mask vector $\boldsymbol{g}$ is directly related to $\beta$ as follows (see Section 3 of main paper for more details):

$$m' = \|G\|_0 = \lceil n(1 + \beta/k) \rceil = m, \tag{17}$$

which means that if we remove the ceiling operator, the following inequality holds:

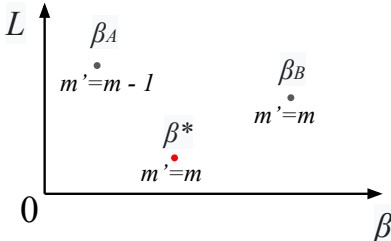

Figure 1: Schematic plot showing variations in the learning objective value $L$ as a function of $\beta$ near the minima of $L$ (shown as red $\beta^*$).

$$m \geq n(1 + \beta/k) > m - 1 \tag{18}$$

$$k\left(\frac{m}{n} - 1\right) \geq \beta > k\left(\frac{m-1}{n} - 1\right) \tag{19}$$

As illustrated in Figure 1, we seek to show that a critical point of $L$ in terms of $\beta$ exists such that $m' = m$. Let $\beta_A = k\left(\frac{m-1}{n} - 1\right), \beta_B = k\left(\frac{m}{n} - 1\right)$ and denote their corresponding values of $L$ as $L_A$ and $L_B$ respectively. As we showed in previous discussions, with $m' \geq m$, as long as $A_1 = \tilde{A_2}^+$, then the first term of $L$ is 0. Thus, there exists a $\beta' < \beta_B$ such that $m' = m$ ($\beta' > \beta_A$), and its corresponding value of $L$ is smaller than $L_B$ (because of a smaller second term). On the other hand, we want to discuss whether the optimization over Eq. (14) will lead to a smaller $\beta$ such that $G$ has more nonzero entries (consequently making the second term of Eq. (14) smaller). In particular, when $m' < m$, then

$$\text{rank}(\hat{X}) = \text{rank}(A_1\tilde{A_2}X) \leq \text{rank}(\tilde{A_2}) \leq \text{rank}(G) \leq m' < m. \tag{20}$$

Since $X$ has no trivial singular values, when $\text{rank}(\hat{X}) < m$, then $\left\|\hat{X} - X\right\|_F^2$ would also take non-trivial values, i.e., $\left\|\hat{X} - X\right\|_F^2 > c\epsilon$, where $c$ is some positive constant. Thus, assuming $\lambda$ is reasonably small, there exists a $\beta' > \beta_A$ such that $m' = m$ ($\beta' \leq \beta_B$), and its corresponding value of $L$ is smaller than $\beta_B$'s corresponding value of $L_A$ (because of a smaller first term).

Thus, there exists a critical point of $L$ (denoted as $\beta^*$) between $\beta_A$ and $\beta_B$, which is the minima of $L$. For $\beta^*$, we have $m' = m$. Therefore we have proven the existence of an optimal solution of DAM. In summary, the optimal set is

$$\Theta^* = \left\{ A_1, A_2, \beta | \forall A_1 \in \mathbb{R}^{d \times n}, \forall A_2 \in \mathbb{R}^{n \times d}, \forall \beta \in \mathbb{R}, \text{s.t.} \right.$$
$$\left. A_1 = (GA_2)^+, k\left(\frac{m}{n} - 1\right) \geq \beta > k\left(\frac{m-1}{n} - 1\right) \right\} \tag{21}$$

## C  Hyperparameter Sensitivity

We have demonstrated that DAM performs very well in the representation learning experiments for dimensionality reduction. Here, we study how stable DAM is with respect to variations in hyperparameter settings. We specifically examine three hyperparameters: learning rate, trade-off parameter, $\lambda$, and the initial value of $\beta$, $\beta_0$. The results of this experiemnt are shown in the form of heatmaps of bottleneck dimension and reconstruction loss upon convergence in Figures 2, 3 and 4.

We can observe that as we vary the learning rate from 0.0001 to 0.1 (which represents a range covering three orders of magnitude), the results for all three cases show variations, where the best results are obtained in the medium range of values (from 0.001 to 0.01), which are common choices of learning rates used in conventional deep learning frameworks. This variation of our results with learning rate is common to any other algorithm based on gradient descent methods. However, for

a fixed learning rate, we can see that the results of DAM are quite stable to the choice of $\lambda$ and $\beta_0$. For example, we can see that the results of DAM are consistent across different choices of cold-start $(\beta^{(0)})$ for all choices of learning rates in the three cases. The sensitivity in terms of $\lambda$ is also low but increases as the feature correlations gets more and more complicated, i.e., as we transition from linear to quadratic to neural network cases. In fact, we observe that small $\lambda$ values result in slower or premature convergence, while too large $\lambda$ can break the model by pruning out all of the parameters. In summary, we can say that by using common choices of learning rate (e.g., 0.01 or 0.001), DAM is able to produce consistent results with reasonable choice of $\lambda$ (e.g., $\lambda = 0.01$).

## D    Experimental Setups

### D.1    Representation Learning (Section 4)

**Datasets:** For dimensionality reduction experiments, we created three synthetic datasets for each of the mapping functions. To generate each dataset, we used two matrices, one as as an encoder and another one as a decoder. For the linear case, the the two matrices were initialized with values sampled from a normal distribution. For the quadratic case, we used a three-layer MLP as the encoder and one QRes layer [1] as the decoder for the quadratic mapping function. For the neural network case, we used a three-layer MLP as the encoder and two-layer MLP as the decoder. We varied the underlying factor ($r$) from 5 to 20 to generate these datasets. For the recommendation system experiments, we used three benchmark datasets Flixter [2], Douban [3], and Yahoo-Music [4]. For graph representation learning, we used three popular citation networks: Cora, CiteSeer, and PubMed [5]. We further used the MNIST [6] dataset to compare between DAM and $L_1$-norm based auto-encoders.

**DAM Implementation Specifications:**

- Table 1 provides full details of the hyper-parameter choices and implementation details of DAM used for generating all of the results in representation learning experiments of Section 4. We used Adam optimizer for all these experiments.

- All our results are reported by taking the mean and standard deviation over five independent random runs.

- We used the linear activation function for the linear dimensionality reduction experiments, LeakyRelu activation function for the quadratic experiments, and the ELU activation function for the neural network mapping experiments. Note that non-linear activation functions were required for the quadratic and neural network experiments, because of the non-linearity of the problem. We added a DAM layer between the encoder and decoder models to perform dimensionality reduction on the bottleneck layer.

- We use the official implementation of IGMC [1] to run the recommendation systems experiments on Flixter, Douban, and Yahoo-Music datasets.

- We use the PyTorch Geometric implementation of GAE [2] for our graph representation learning experiments.

- We kept the same architecture as the official implementations of IGMC and GAE for all the recommendation system and graph representation learning experiments. For IGMC-DAM and GAE-DAM, we simply added one DAM layer after the bottleneck layer to perform pruning of the bottleneck dimension.

- For the dimensionality reduction experiments on the MNIST dataset, we used a plain autoencoder setup. We used four layers with 784, 64, 32, 50 neurons for the encoder, and for the decoder, we used four layers with 50, 32, 64, and 784 neurons. ReLU was used as the activation function. After the encoder layer, we used a DAM layer for dimensionality reduction. Likewise, we append an $L_1$-norm layer after the encoder layer for the $L_1$-norm-based dimensionality reduction.

- We varied $\lambda$ values for the graph representation learning experiments and MNIST dataset as follows. We varied the $\lambda$ from 0.01 to 5 for both GAE and GAE-DAM for graph

---

[1]`https://github.com/muhanzhang/IGMC`
[2]`https://github.com/rusty1s/pytorch_geometric/blob/master/examples/autoencoder.py`

Table 1: Experiment Setups And Implementation Details For Representation Learning (Section 4).

| Source | Dataset | Model | Learning Rate | Epochs | $\lambda$ | Cold-Start |
|--------|---------|-------|---------------|--------|-----------|------------|
| Fig. 3a | Synthetic | Linear | $0.01,\ L_2 = 10^{-6}$ | 2,000 | 0.01 | $\beta^{(0)} = 1,\ 0$ epochs |
| Fig. 3b | Synthetic | Quadratic | $0.01,\ L_2 = 10^{-6}$ | 5,000 | 0.01 | $\beta^{(0)} = 5,\ 0$ epochs |
| Fig. 3c | Synthetic | NN | $0.001,\ L_2 = 0$ | 10,000 | 0.1 | $\beta^{(0)} = 1,\ 0$ epochs |
| Tab. 2 | Flixter | IGMC | $0.001,\ L_2 = 0$ | 100 | - | - |
| Tab. 2 | Flixter | IGMC-DAM | $0.001,\ L_2 = 0$ | 100 | 0.1 | $\beta^{(0)} = 1,\ 0$ epochs |
| Tab. 2 | Douban | IGMC | $0.001,\ L_2 = 0$ | 100 | - | - |
| Tab. 2 | Douban | IGMC-DAM | $0.001,\ L_2 = 0$ | 100 | 0.1 | $\beta^{(0)} = 1,\ 0$ epochs |
| Tab. 2 | Yahoo-Music | IGMC | $0.001,\ L_2 = 0$ | 100 | - | - |
| Tab. 2 | Yahoo-Music | IGMC-DAM | $0.001,\ L_2 = 0$ | 100 | 50.0 | $\beta^{(0)} = 1,\ 0$ epochs |
| Fig. 4a | Cora | GAE-DAM | $0.01,\ L_2 = 0$ | 1,000 | variable | $\beta^{(0)} = 1,\ 0$ epochs |
| Fig. 4a | CiteSeer | GAE-DAM | $0.01,\ L_2 = 0$ | 1,000 | variable | $\beta^{(0)} = 1,\ 0$ epochs |
| Fig. 4a | PubMed | GAE-DAM | $0.01,\ L_2 = 0$ | 1,000 | variable | $\beta^{(0)} = 1,\ 0$ epochs |
| Fig. 5 | MNIST | $L_1$-norm | $0.001,\ L_2 = 0$ | 100 | variable | - |
| Fig. 5 | MNIST | DAM | $0.001,\ L_2 = 0$ | 100 | variable | $\beta^{(0)} = 1,\ 0$ epochs |

> representation learning experiments. We varied the $\lambda$ from 0.01 to 10 for the DAM model on the MNIST dataset. Likewise, we changed the range of $\lambda$ from 0.01 to 25 for the $L_1$-norm model.

- We observed some improvement in the performance of the baseline IGMC model for 100 epochs, although the official implementation used 40 epochs. We kept all other hyperparameters same as the official implementation.

### D.2 Structured Network Pruning (Section 5)

**DAM Implementation Specifications:**

- All the models were trained using SGD optimizer.

- Table 2 provides full details of the hyperparameter choices and implementation details of DAM for all structured network pruning experiments.

- For running time comparisons, we used an *NVIDIA TITAN RTX* graphic card with *Intel Xeon Gold 6240* CPU on *Ubuntu 18.04 LTS* system. The channel pruning ratio for Net-Slim and ChipNet is set to 0.4 in all our experiments.

- For pruning experiments, we used learning rate decay as adopted in the ChipNet official implementation[3]. Table 2 only shows the initial learning rate in these experiments.

- For all methods, the epochs are shown in the format of training epochs + pruning epochs + finetuning epochs. We can see that DAM requires 0 pruning epochs and 0 finetuning epochs, Net-Slim has 0 finetuning epochs, whereas ChipNet has non-zero epochs for all three stages.

- For Figure 6, on CIFAR-10 we use the following range of $\lambda$ values for DAM, $\lambda = \{0.0, 0.1, 0.2, 0.3, 0.4, 0.5, 0.75\}$. For ChipNet and Net-Slim, we used the following range of values for pruning ratios: $\{0.1, 0.2, 0.4, 0.6, 1.0\}$ and $\{0.1, 0.2, 0.3, 0.4, 0.5, 0.6, 1.0\}$, respectively. For CIFAR-100, we use $\lambda = \{0.1, 0.2, 0.3, 0.4, 0.5, 0.75, 1.0\}$ for DAM, and pruning ratios of $\{0.2, 0.4, 0.6, 1.0\}$ and $\{0.2, 0.3, 0.4, 0.5, 0.6, 1.0\}$ for ChipNet and Network Slimming, respectively. We did not report the 0.1 pruning ratio on CIFAR-100 since both ChipNet and Network Slimming demonstrated unstable results.

## E Additional Experimental Results

### E.1 Effect of Gradient Noise and Activation Functions on MNIST Dataset

We performed further experiments to evaluate the network pruning performance of DAM using LeNet-5 on MNIST dataset, which is a common dataset for experiments adopted by many previous

---

[3]`https://github.com/transmuteAI/ChipNet`

Table 2: Experiment Setups For Structured Network Pruning (Section 5).

| Source | Dataset | Model | Learning Rate | Epochs | $\lambda$ | Cold-Start |
|---|---|---|---|---|---|---|
| Fig. 6 | CIFAR-10 | DAM | 0.05, $L_2 = 10^{-3}$ | 200+0+0 | variable | $\beta^{(0)} = 1$, 20 epochs |
| Fig. 6 | CIFAR-10 | Net-Slim | 0.05, $L_2 = 10^{-3}$ | 200+0+50 | - | - |
| Fig. 6 | CIFAR-10 | ChipNet | 0.05, $L_2 = 10^{-3}$ | 200+20+50 | - | - |
| Fig. 6 | CIFAR-100 | DAM | 0.05, $L_2 = 10^{-3}$ | 200+0+0 | variable | $\beta^{(0)} = 1$, 20 epochs |
| Fig. 6 | CIFAR-100 | Net-Slim | 0.05, $L_2 = 10^{-3}$ | 200+0+50 | - | - |
| Fig. 6 | CIFAR-100 | ChipNet | 0.05, $L_2 = 10^{-3}$ | 200+20+50 | - | - |
| Fig. 7 | CIFAR-10 | DAM | 0.05, $L_2 = 10^{-3}$ | 200+0+0 | 0.4 | $\beta^{(0)} = 1$, 20 epochs |
| Fig. 7 | CIFAR-10 | Net-Slim | 0.05, $L_2 = 10^{-3}$ | 200+0+50 | - | - |
| Fig. 7 | CIFAR-10 | ChipNet | 0.05, $L_2 = 10^{-3}$ | 200+20+50 | - | - |
| Tab. 2 | CIFAR-10 | DAM | 0.05, $L_2 = 10^{-3}$ | 200+0+0 | 0.4 | $\beta^{(0)} = 1$, 20 epochs |
| Tab. 2 | CIFAR-10 | Net-Slim | 0.05, $L_2 = 10^{-3}$ | 200+0+50 | - | - |
| Tab. 2 | CIFAR-10 | ChipNet | 0.05, $L_2 = 10^{-3}$ | 200+20+50 | - | - |

works. Table 3 (a) shows the results of DAM for pruning the LeNet model at varying values of $\lambda$. Figure 5 also shows how test accuracy and pruning ratio varies as we change $\lambda$. We can see that the pruning keeps on continuing even with aggressively large values of $\lambda$ (close to 2), which again demonstrates that DAM does not suffer from the saturation effects of $L_1$-based regularization, as was described in Section 4 of the main paper. In addition to pruning the LeNet, we also examine how adding gradient noise [7] at every epoch of neural network training affects the results. Table 3 (b) shows that by adding gradient noises to the training process leads to slight improvements in accuracy and pruning fractions, implying that more stochasticity in the training process may improve the ability of DAM to perform network pruning. Table 3 (c) further shows the results of DAM on LeNet-5, where we replace all the activation functions with the sine activation function as proposed in [8]. Interestingly, the results are much better with sine activation function, implying that the DAM may be compatible with periodic activation functions.

## E.2 Additional Results on CIFAR Datasets

Table 4 presents additional results of DAM on CIFAR datsets for many other neural network architectures than what was shown in the main paper, including VGG-19, PreResnet-20, PreResNet-56, and PreResnet-110. Note that the DAM results presented in the main paper were obtained using an implementation of our algorithm based on the ChipNet source code [9], which was published just a few weeks before the time of our submission. Given the recentness of this implementation, we were not able to complete extensive evaluations on all architectures using this implementation. Instead, the results presented in this section are based on an alternate implementation of our DAM algorithm based on the stable source code provided in an older well-established previous work [10]. In this implementation, the DAM is trained for 160 epochs using AdaBelief [11] optimizer. We release both implementations of our DAM algorithms in the anonymous link of the source code provided in the main paper.

## E.3 Analyzing Similarity in Pruned Features

A useful property of *compact* features learned at the hidden layers of a neural network is that they show express 'distinct' features upon convergence, such that pruning the neurons any further would lead to drop in accuracy. To evaluate the similarity in the features extracted by comparative structured network pruning methods, we compute the centered kernel alignment (CKA) similarity [12] matrix for all pairs of neurons with non-zero mask values at the 54-th layer in the PreResNet-164 architecture pruned using DAM, Net-Slim, and ChipNet. Note that we chose the 54-th layer as it represents one third of the total number of layers at the end of the first 'BaseBlock' of PreResNet-164 for ease of implementation, although these results can be visualized for any other layer number too. The CKA similarity matrices of the unpruned network and the three pruned networks are shown in Figure 6, where higher off-diagonal values in these matrices represent higher similarity among the features. To further quantify the differences between DAM and the baseline methods, we compute the statistics of the values in the off-diagonal elements of the CKA matrices in Table 5. We can observe observe that DAM shows lowest CKA similarity on the off-diagonal elements as compared to Net-Slim and

Table 3: DAM on MNIST Dataset, LeNet-5 (# Neurons 6-16-120)

(a) Tanh Activation w.o. Gradient Noise

| $\lambda$ | 0 | 0.01 | 0.02 | 0.05 | 0.1 | 0.2 | 0.3 | 0.4 | 0.5 | 1.0 |
|---|---|---|---|---|---|---|---|---|---|---|
| Parameters | 60.0k | 33.6k | 18.7k | 7.0k | 3.9k | 1.9k | 1.4k | 1.3k | 1.0k | 0.7k |
| Pruned (%) | 0.00 | 44.15 | 68.82 | 88.41 | 93.59 | 96.78 | 97.71 | 97.85 | 98.31 | 98.90 |
| Accuracy (%) | 98.85 | 98.61 | 98.47 | 98.16 | 97.49 | 96.54 | 95.47 | 94.79 | 93.70 | 90.12 |

(b) Tanh Activation with 5% Gradient Noise

| $\lambda$ | 0 | 0.01 | 0.02 | 0.05 | 0.1 | 0.2 | 0.3 | 0.4 | 0.5 | 1.0 |
|---|---|---|---|---|---|---|---|---|---|---|
| Parameters | 60.0k | 35.0k | 18.8k | 5.9k | 3.9k | 1.8k | 1.4k | 1.3k | 1.0k | 0.6k |
| Pruned (%) | 0.00 | 41.71 | 68.70 | 90.19 | 93.53 | 96.97 | 97.67 | 97.85 | 98.31 | 99.00 |
| Accuracy (%) | 98.78 | 98.57 | 98.69 | 98.03 | 97.69 | 96.61 | 95.64 | 94.78 | 94.16 | 90.16 |

(c) Sine Activation w.o. Gradient Noise

| $\lambda$ | 0 | 0.01 | 0.02 | 0.05 | 0.1 | 0.2 | 0.3 | 0.4 | 0.5 | 0.6 |
|---|---|---|---|---|---|---|---|---|---|---|
| Parameters | 60.0k | 21.5k | 9.2k | 3.4k | 1.9k | 1.5k | 1.1k | 1.2k | 0.8k | 0.6k |
| Pruned (%) | 0.00 | 64.24 | 84.64 | 94.27 | 96.84 | 97.44 | 98.12 | 98.05 | 98.59 | 98.95 |
| Accuracy (%) | 98.93 | 99.00 | 98.85 | 98.53 | 98.38 | 97.12 | 96.12 | 96.63 | 94.91 | 95.49 |

ChipNet, indicating that the features learned by DAM are more distinct from one another. Also note that in Figure 6, the size of the pruned features (with non-zero values) extracted by DAM is quite smaller than what we obtain from Net-Slim and ChipNet using their standard implementations made available by their authors. This is because some proportion of pruned channels become nonzero after finetuning in Net-Slim (see details of mask implementation here[4]) and ChipNet. Their current implementations thus result in smaller actual pruning ratios after finetuning for practical use than what is reported (before finetuninng).

### E.4 Visualization of Pruned Architectures

**PreResNet-164**: Figures 7 and 8 show the visualizations of architectures used in Section 5 of the main paper. For small trade-off parameters $\lambda$, the DAM tends to prune evenly across layers. For extreme pruning ratios ($\lambda = 1.0$), the DAM prunes a lot of blocks, showing that it is more in favor of widths than depths (which happens to agree with the idea of *WideResNet* [13]).

**Permutation Invariance**: In order to further support the permutation invariance of DAM (reported in the main paper), we plot the architectures of PreResNet-164 pruned using DAM after randomly permuting the neuron ordering. We can see from Figure 9 that the resulting architectures are consistent across different permutations.

**Hinged Variant of DAM**: Most of the pruning algorithms result in uneven pruning across different layers which may create difficulty for pruning over the sparsified models, especially for models with shortcut connections like ResNet and DenseNet. To show that DAM is easy to use and easily customizable according to the needs of the base network architecture, we present a Hinged variant of DAM, where the major difference is that the bottleneck dimensions of the ResNet are hinged to be the same. The pruned architectures of PreResNet-20, PreResNet-56 and PreResNet-110 are shown in Figures 10 and 11, where we can see that the Hinged variant of DAM produces comparable results as the normal DAM.

---

[4]https://github.com/Eric-mingjie/network-slimming/tree/master/mask-impl

Table 4: Additional results of pruning different architectures using DAM on CIFAR datasets.

| Network | Dataset | λ | Top-1 (%) | Params (k) | Pruned C. (%) | Pruned P. (%) |
|---|---|---|---|---|---|---|
| VGG-19 | CIFAR-10 | 0.1 | 93.39 | 1,627 | 71.28 | 91.88 |
| VGG-19 | CIFAR-10 | 0.5 | 90.92 | 264 | 89.17 | 98.68 |
| VGG-19 | CIFAR-100 | 0.1 | 73.40 | 6,190 | 41.30 | 69.17 |
| VGG-19 | CIFAR-100 | 0.5 | 67.17 | 664 | 79.81 | 96.69 |
| PreResNet-20 | CIFAR-100 | 0.1 | 67.33 | 228 | 2.72 | 5.88 |
| PreResNet-20 | CIFAR-100 | 0.2 | 66.53 | 217 | 5.37 | 10.50 |
| PreResNet-20 | CIFAR-100 | 0.3 | 66.46 | 206 | 7.72 | 15.13 |
| PreResNet-20 | CIFAR-100 | 0.4 | 65.83 | 191 | 11.25 | 21.18 |
| PreResNet-20 | CIFAR-100 | 0.5 | 66.17 | 183 | 14.12 | 24.67 |
| PreResNet-20 | CIFAR-100 | 1.0 | 63.15 | 121 | 31.84 | 50.05 |
| PreResNet-20 | CIFAR-100 | 2.0 | 54.79 | 47 | 62.35 | 80.45 |
| PreResNet-56 | CIFAR-100 | 0.1 | 72.74 | 573 | 3.33 | 6.60 |
| PreResNet-56 | CIFAR-100 | 0.2 | 72.64 | 526 | 7.93 | 14.35 |
| PreResNet-56 | CIFAR-100 | 0.3 | 72.49 | 493 | 10.72 | 19.67 |
| PreResNet-56 | CIFAR-100 | 0.4 | 72.28 | 466 | 14.62 | 24.14 |
| PreResNet-56 | CIFAR-100 | 0.5 | 71.73 | 419 | 19.17 | 31.73 |
| PreResNet-56 | CIFAR-100 | 1.0 | 69.94 | 314 | 33.60 | 48.77 |
| PreResNet-56 | CIFAR-100 | 2.0 | 64.31 | 153 | 62.94 | 75.05 |
| PreResNet-110 | CIFAR-10 | 0.1 | 94.51 | 975 | 9.72 | 15.01 |
| PreResNet-110 | CIFAR-10 | 0.2 | 93.73 | 808 | 20.71 | 29.54 |
| PreResNet-110 | CIFAR-10 | 0.3 | 93.28 | 646 | 31.10 | 43.65 |
| PreResNet-110 | CIFAR-10 | 0.4 | 93.35 | 560 | 37.08 | 51.18 |
| PreResNet-110 | CIFAR-10 | 0.5 | 92.97 | 482 | 46.55 | 58.00 |
| PreResNet-110 | CIFAR-10 | 1.0 | 91.61 | 208 | 72.51 | 81.85 |
| PreResNet-110 | CIFAR-10 | 2.0 | 88.90 | 89 | 84.99 | 92.21 |
| PreResNet-110 | CIFAR-100 | 0.1 | 75.55 | 1110 | 4.20 | 5.15 |
| PreResNet-110 | CIFAR-100 | 0.2 | 74.73 | 1010 | 10.24 | 13.70 |
| PreResNet-110 | CIFAR-100 | 0.3 | 75.08 | 940 | 14.54 | 19.63 |
| PreResNet-110 | CIFAR-100 | 0.4 | 73.93 | 873 | 18.37 | 25.41 |
| PreResNet-110 | CIFAR-100 | 0.5 | 73.44 | 817 | 21.87 | 30.17 |
| PreResNet-110 | CIFAR-100 | 1.0 | 71.63 | 547 | 39.68 | 53.21 |
| PreResNet-110 | CIFAR-100 | 2.0 | 68.32 | 261 | 66.45 | 77.72 |

Table 5: Statistics of CKA similarity between different features learned at layer 54 of PreResNet-164 after pruning (calculated using the off-diagonal elements in Figure 6).

| CKA | Unpruned | DAM | Net-Slim | ChipNet |
|---|---|---|---|---|
| mean | 0.273 | **0.229** | 0.328 | 0.295 |
| std | 0.018 | **0.013** | 0.021 | 0.020 |
| max | 0.840 | **0.740** | 0.893 | 0.964 |

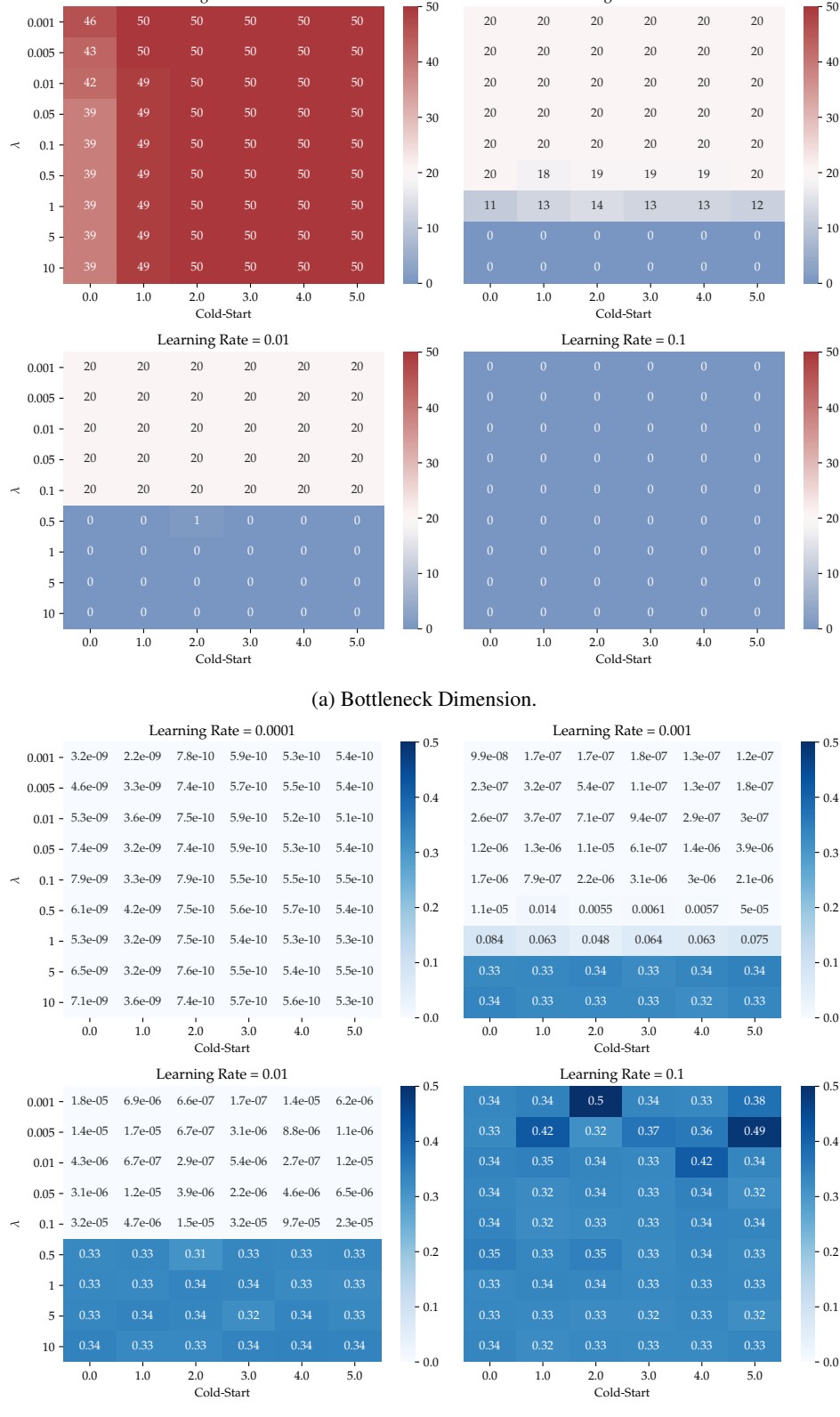

(a) Bottleneck Dimension.

(b) Reconstruction Loss.

Figure 2: Hyper-parameter sensitivity of DAM for dimensionality reduction (Section 4, Linear).

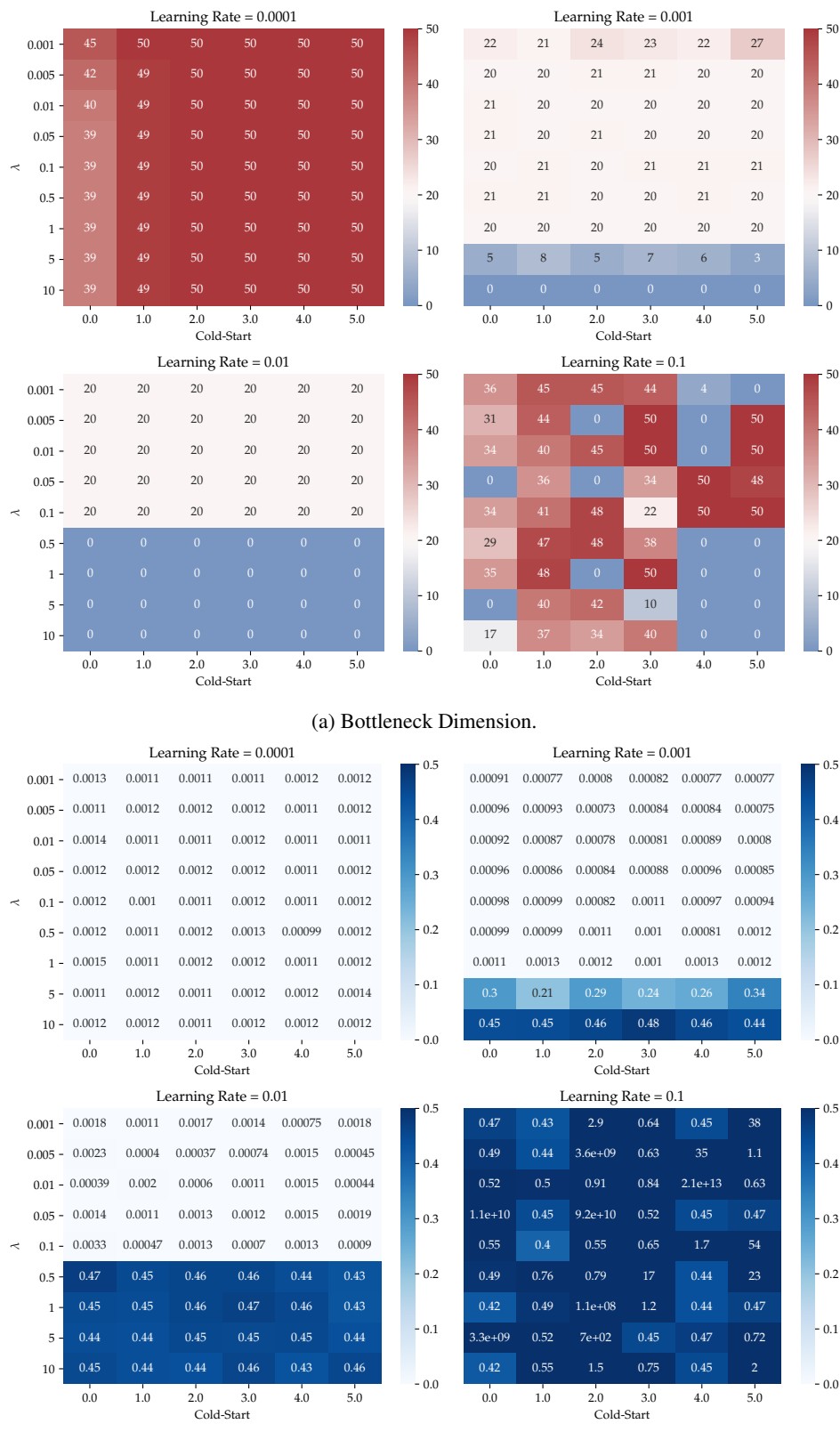

(a) Bottleneck Dimension.

(b) Reconstruction Loss.

Figure 3: Hyper-parameter sensitivity of DAM for dimensionality reduction (Section 4, Polynomial).

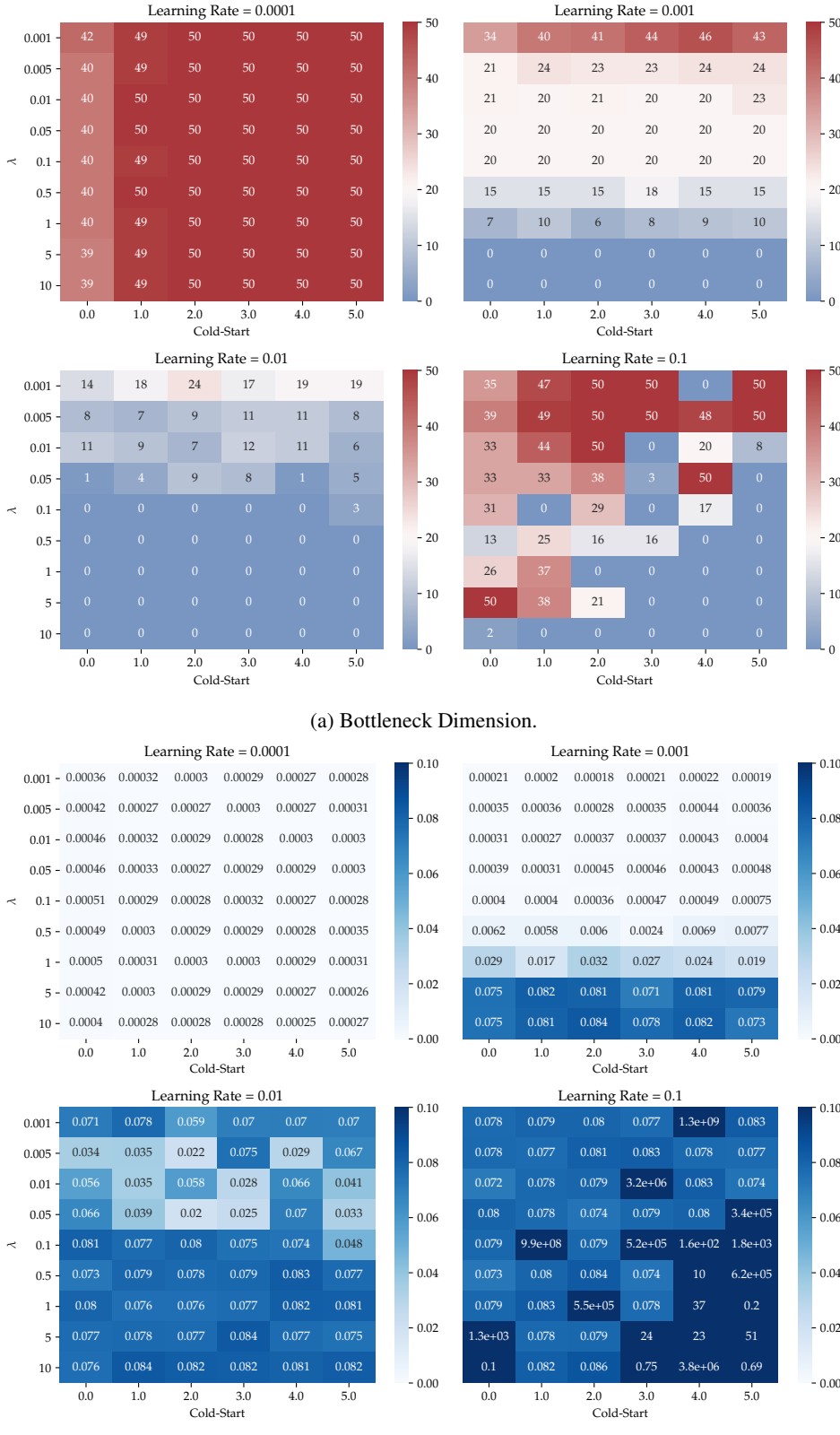

(a) Bottleneck Dimension.

(b) Reconstruction Loss.

Figure 4: Hyper-parameter sensitivity of DAM for dimensionality reduction (Section 4, Neural Network).

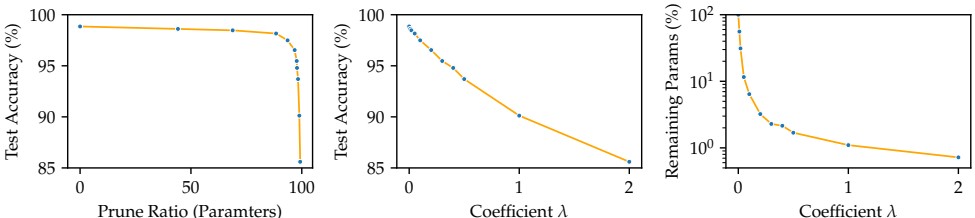

Figure 5: Results of DAM trained with varying $\lambda$ on MNIST dataset. Accuracy is reported using the mean values of 5 random runs.

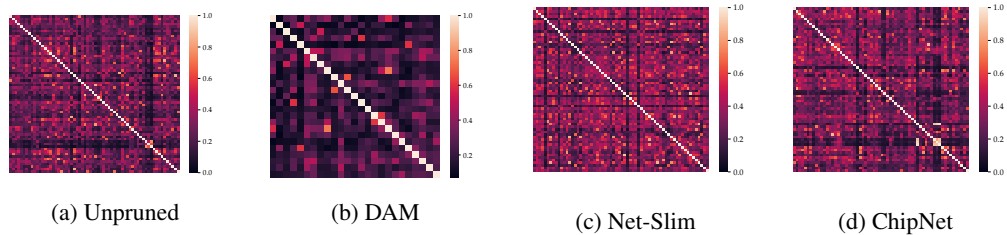

(a) Unpruned     (b) DAM     (c) Net-Slim     (d) ChipNet

Figure 6: CKA similarity between features learned at layer 54 of PreResNet-164 model, before pruning, and after pruning using the three methods.

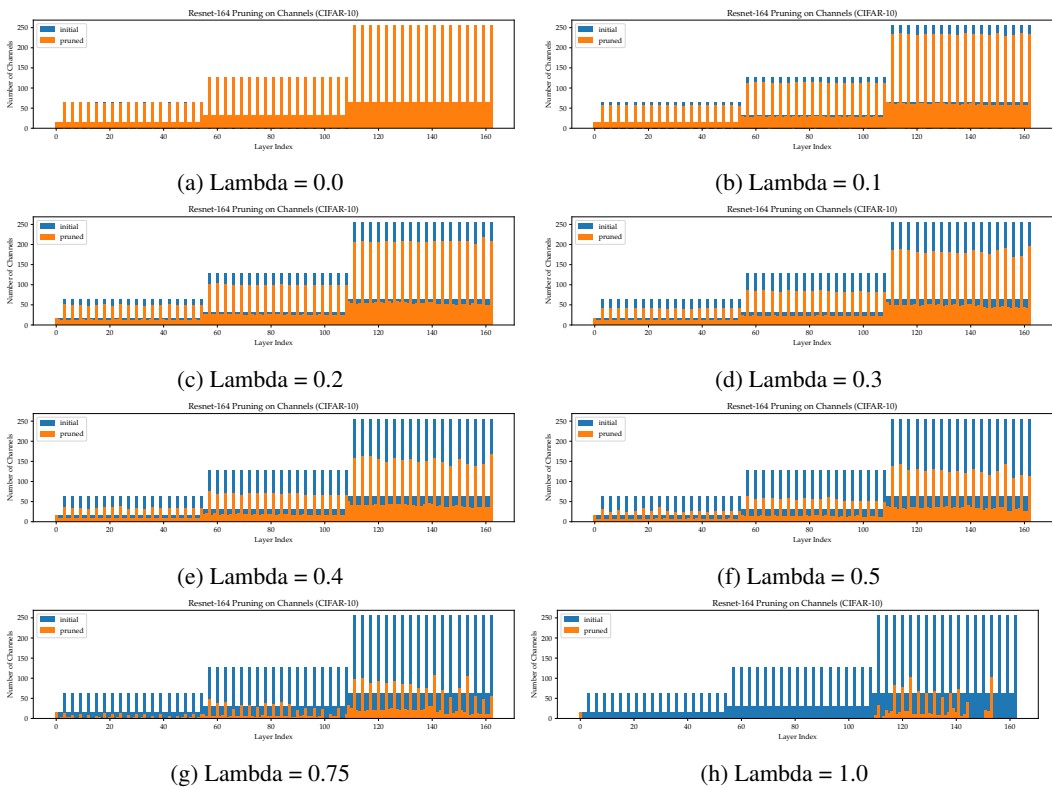

(a) Lambda = 0.0       (b) Lambda = 0.1

(c) Lambda = 0.2       (d) Lambda = 0.3

(e) Lambda = 0.4       (f) Lambda = 0.5

(g) Lambda = 0.75       (h) Lambda = 1.0

Figure 7: Visualization of PreResNet-164 Pruned on CIFAR-10

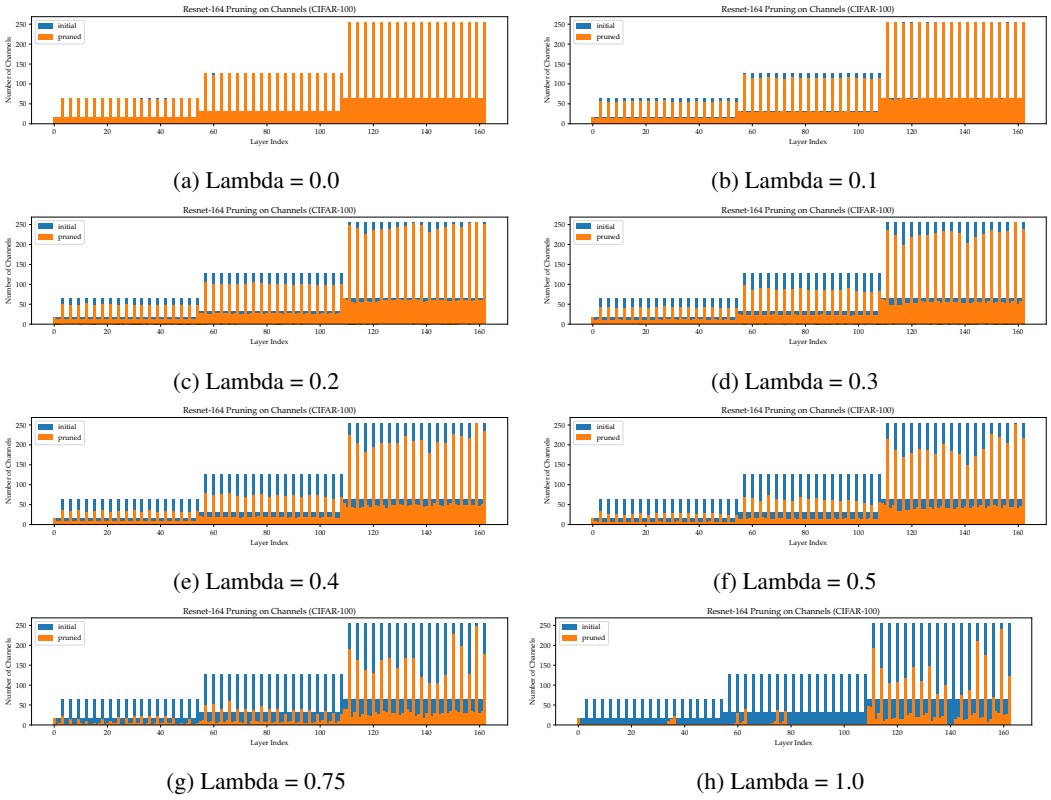

(a) Lambda = 0.0

(b) Lambda = 0.1

(c) Lambda = 0.2

(d) Lambda = 0.3

(e) Lambda = 0.4

(f) Lambda = 0.5

(g) Lambda = 0.75

(h) Lambda = 1.0

Figure 8: Visualization of PreResNet-164 Pruned on CIFAR-100

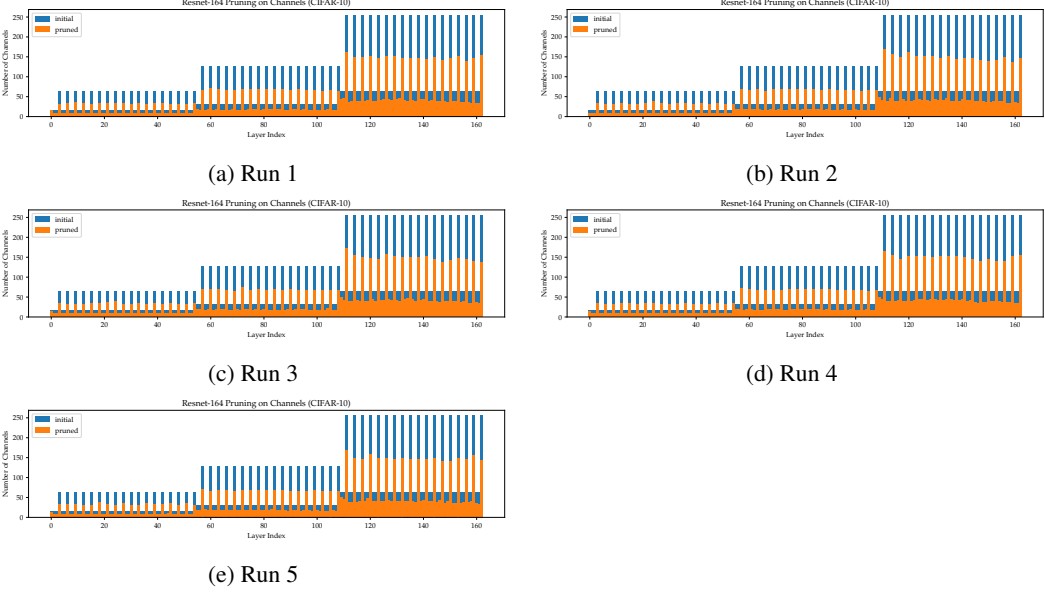

(a) Run 1

(b) Run 2

(c) Run 3

(d) Run 4

(e) Run 5

Figure 9: Permutation Invariance Experiment (Table 2 in Section 5) on CIFAR-10

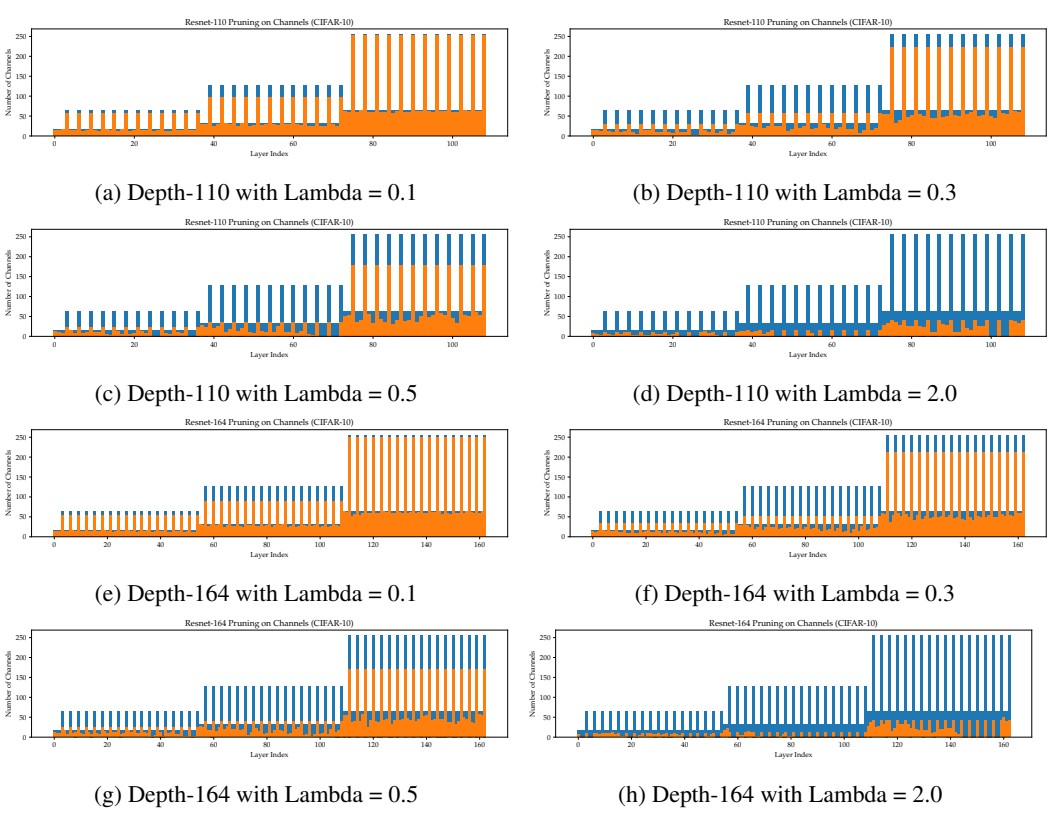

(a) Depth-110 with Lambda = 0.1

(b) Depth-110 with Lambda = 0.3

(c) Depth-110 with Lambda = 0.5

(d) Depth-110 with Lambda = 2.0

(e) Depth-164 with Lambda = 0.1

(f) Depth-164 with Lambda = 0.3

(g) Depth-164 with Lambda = 0.5

(h) Depth-164 with Lambda = 2.0

Figure 10: Hinged-Variant of various PreResNet architectures Pruned on CIFAR-10

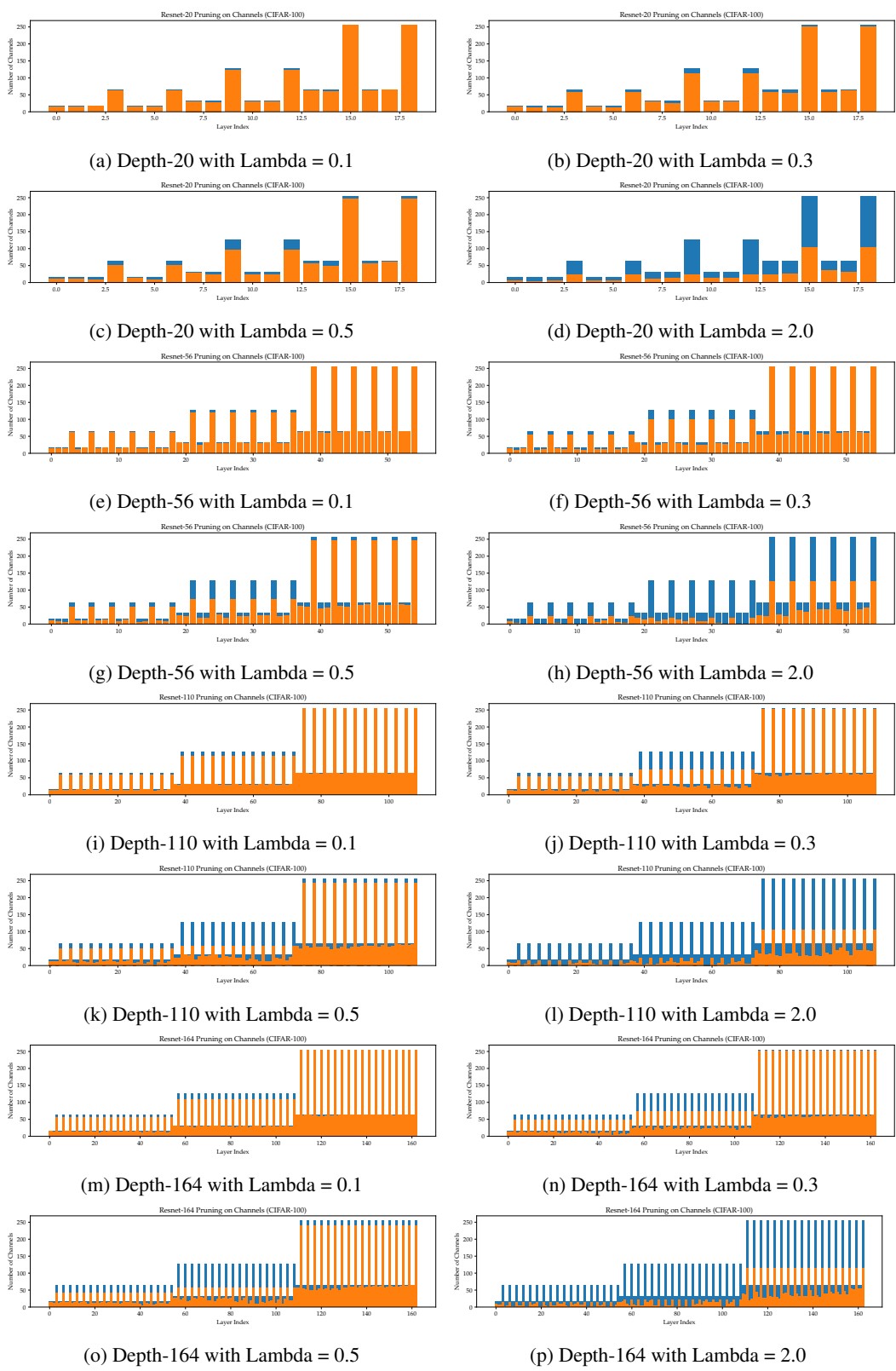

Figure 11: Hinged-Variant of various PreResNet architectures Pruned on CIFAR-100