# OpenReview forum: "Learning Compact Representations of Neural Networks using DiscriminAtive Masking (DAM)"
_NeurIPS.cc/2021/Conference — NeurIPS 2021 Poster_

### Official Review · Reviewer_uBCs · 2021-07-05

**Rating:** 4
**Confidence:** 3

**Summary:**

The paper proposes a single-stage structured pruning method, DiscriminAtive Masking (DAM), which learns a mask to discriminatively identify neurons that needs to be pruned out.

**Limitations And Societal Impact:**

Yes.

**Main Review:**

Overall, the contribution of the paper is not that clear to me. My first impression is that this paper proposes a structured pruning method (as indicated in the abstract) and expect that it will conduct extensive experiments on pruning/compressing various networks on various benchmark datasets and compare the performance (generalization/FLOPS/runtime/size) with other state-of-the-art methods. However, it spends 2-3 pages in describing its applications in dimensionality reduction, recommendation system and graph representation learning, which confused me a lot. Most importantly, due to the limited length of the paper and so many topics it intends to cover, for each of these applications as well as the structured pruning in Section 5, the demonstrated experiments are rather superficial: either only synthetic data is involved in the experiments or the algorithm is only evaluated on toy dataset (MNIST, CIFAR10...). In Section 5 of structured pruning, the competitors are only restricted to [15,16], which is not that convincing to me.

I would recommend concentrating on one direction, namely the structured pruning, and conducting serious experiments on this particular part. For example, it would be better to test different backbones, e.g., VGG/ResNet/DenseNet, on ImageNet, and compare the FLOPS reduction, generalization performance, inference speed-up of the resulting compressed model with many other state-of-the-art methods. This would highlight the main contribution to a much larger extent than the current practice.

In the same time, I think the paper lacks a basic proofreading since there are so many typos or mistakes:

- line 73, function "g" should be bold to be consistent with the notation later?
- equation 1, the subscript g in "F_g" should also be bold; the "x" in "x\sim ..." should be bold
- line 152, \theta is not only the learnable parameters of the network, since it also optimizes over \beta. The current formulation is a little bit wired since there is no \theta appears in the objective function
- Figure 2, in the caption "k=... \beta=..." should be in math mode when typesetting; in (c), (d), what is the setting for \alpha and \mu?
- line 171-172, it says the learnable \beta introduces "a tremendous amount of simplicity", which is a negative description, however, what it actually wants to convey is that this is a positive aspect of the proposed method
- line 207, after the bold phrase, there should be a period at the end
- line 268, after "Lasso)" there are two periods
- line 304, the "E[\beta]" should be in math mode; more importantly, I think think the plot is for the expectation of \beta, since it is impossible to collect infinitely many of running samples for calculating such expectation, what you actually do here is only the average of \beta?







**Time Spent Reviewing:**

5

---

> ### Author Response · Authors · 2021-08-11
> **Author Response to Reviewer uBCs**
>
> We are very grateful to the reviewer for their valuable feedback. Detailed responses to the comments are provided below.
>
> ***Comment 1***
>
> Overall, the contribution of the paper is not that clear to me. My first impression is that this paper proposes a structured pruning method … However, it spends 2-3 pages in describing its applications in dimensionality reduction, recommendation system and graph representation learning, which confused me a lot. I would recommend concentrating on one direction, namely the structured pruning, and conducting serious experiments on this particular part.
>
> **Response**
>
> We consider our joint analysis of the two problems of representation learning and structured network pruning under the common umbrella of “learning compact representations” as one of the major strengths of our work. While we could have chosen to solely focus on structured network pruning experiments, we believe this would have limited the scope and impact of our work. Instead, we chose to provide a unified perspective of these two seemingly different but related problems, which we hope motivates future research to further bridge their gaps and explore the advantages of learning compact representations on parameter efficiency and generalizability for both problems. We emphasize our unified perspective at multiple places in the paper, including the introduction section and the problem statement covering both problems (see Figure 2). We will further bolster this point in the abstract to clarify our joint focus on both these problems.
>
> Another motivation behind covering both representation learning and structured pruning in our work is the richness and diversity of theoretical and empirical explorations enabled by their joint perspective. For example, our experiments on dimensionality reduction on synthetic datasets provide the foundation for our theoretical analyses of DAM presented in Appendix B, where we prove the convergence of DAM to the optimal solution for the linear dimensionality reduction case. Our results in graph representation learning demonstrate the ability of DAM to achieve state-of-the-art (SOTA) performance even with a very simple choice of backbone network. Our results on MNIST data demonstrate the ability of DAM to avoid saturation in pruning even at large values of lambda, in contrast to L1 based methods. We also perform extensive experiments in structured network pruning to demonstrate the efficacy of DAM with respect to cutting-edge SOTA methods (published a few weeks before NeurIPS deadline) in terms of accuracy, training time, and training stability over standard benchmark datasets and backbone architectures used in previous literature. This variety of investigations helps us evaluate the strengths of our proposed approach from multiple perspectives, which we consider to be important for a novel methodology such as DAM.
>
> ***Comment 2***
>
> The demonstrated experiments are rather superficial: either only synthetic data is involved in the experiments or the algorithm is only evaluated on toy dataset (MNIST, CIFAR10...)
>
> ***Response***
>
> We were motivated to include results on synthetic data for dimensionality reduction as our first line of evidence so as to have full control on the exact number of latent dimensions and the amount of non-linearity in the latent embeddings of the simulated data. This helped in testing the ability of DAM to recover the latent representations in a controlled setting, and also served as the foundation for our theoretical analysis proving the ability of DAM to converge to the optimal solution in the linear dimensionality reduction case, as presented in Appendix B. The motivation behind using MNIST was to empirically demonstrate the ability of DAM to avoid the problem of saturation while pruning at large values of lambda, in contrast to L1 based methods. Finally, our motivation behind using CIFAR datasets was to be consistent with previous literature in structured network pruning that have used them as benchmark datasets (e.g., ChipNet). In particular, we chose CIFAR datasets so that we don’t have to finetune baseline methods such as ChipNet on new data but can rather use the tuned code already provided by the official implementations of these methods.
>
> ***Comment 3***
>
> In Section 5 of structured pruning, the competitors are only restricted to [15,16], which is not that convincing to me. It would be better to test different backbones, e.g., VGG/ResNet/DenseNet, on ImageNet, and compare the FLOPS reduction, generalization performance, inference speed-up of the resulting compressed model with many other SOTA methods. This would highlight the main contribution to a much larger extent than the current practice.
>
> ***Response***
>
> This is a very helpful comment. In the following, we describe our motivation behind the choice of baselines, backbone networks, datasets, and evaluation procedures. We also present our new results on TinyImageNet going beyond CIFAR datasets.
>
> **Choice of Baselines**: We chose ChipNet as one of the primary baselines in our work, which represents the latest SOTA in structured network pruning published in May 2021 (a few weeks before the NeurIPS deadline).  The original paper of ChipNet included extensive evaluations with a large number of baselines in structured network pruning on CIFAR datasets, where it was shown that ChipNet significantly outperforms all previous baselines. Hence, by showing comparable accuracy as Chipnet but with better parameter efficiency, training speed, and training stability, we are able to demonstrate that DAM achieves SOTA performance on benchmark datasets. We used the official Implementation of ChipNet (https://github.com/transmuteAI/ChipNet) to have fair comparisons in all our experiments. Another baseline that we considered is Network Simming, which is one of the earlier and well-established methods in structured pruning that is known to be a  strong competitor in previous works.
>
> **Choice of Backbone Networks**: Given the time and space constraints, we could not include results exhaustively over all recent network architectures (e.g., DenseNets) but only those that have been commonly used in existing literature on structured network pruning, including VGG-19, PreResnet-20, PreResNet-56, PreResnet-110, and PreResNet-164 on CIFAR datasets and LeNet-5 on MNIST (see Appendix E3 for additional results). To be consistent with existing literature on structured network pruning (e.g., ChipNet and Network Slimming), we used PreResNet-164 as the primary backbone network in our analysis in Section 5. Also note that PreResNet-164 is a relatively compact model that only has around 1M parameters, making it non-trivial to prune the network further.
>
> **Evaluation on Larger Datasets**: While CIFAR datasets represent a standard benchmark for evaluating structured pruning methods from scratch, here we present some new results on larger datasets going beyond CIFAR. Note that training on ImageNet requires significant computation time and hardware resources that are difficult to obtain in academic settings, which is one of the reasons why it has not been adopted in SOTA evaluations (e.g., ChipNet). Instead, we were able to generate some new results for the Tiny-ImageNet (http://cs231n.stanford.edu/reports/2017/pdfs/930.pdf) in response to the reviewer comments. For this dataset, we used PreResNet-164 as the backbone network with standard configurations of DAM as presented in Section 5 of the main paper (with additional details in Appendix D.2). We use the AdaBelief optimizer for faster convergence on this larger dataset.
>
> We trained the models from scratch for 200 epochs and obtained the following results:
>
> |  Lambda  | Top-1 Test Accuracy |Channels pruned (%) |
> |:--:|:--:|:--:|
> |Unpruned |52.72 | 0.00 |
> | 0.1 | 52.75   | 6.03 |
> | 0.3 | 52.51 | 20.05 |
> | 0.5 | 52.99 | 30.91 |
> | 0.7 | 52.11 | 38.71 |
> | 1.0 | 52.89 | 45.94 |
> | 2.0 | 53.17 | 68.52 |
>
> We can see that DAM is able to consistently maintain the test accuracy on TinyImageNet even at extreme levels of network pruning close to 68%. This demonstrates the ability of DAM to achieve useful generalization on larger datasets going beyond CIFAR. Note that these results were generated without any data augmentations, label smoothings, transfer learning, and other advanced procedures employed by SOTA methods developed for TinyImageNet, which explains the lower range of test accuracies in the above table. However, the goal of our experiment was to only demonstrate the ability of DAM to perform pruning along with maintaining test accuracy on larger datasets, instead of achieving SOTA performance with the most advanced architectures. We will include these results in the Supplementary Materials of the paper.
>
> **Choice of Evaluation Procedures**: Along with comparing results in terms of test accuracy, training time, and parameter efficiency (measured in terms of number of parameters pruned), we also analyze the similarity between the learned features of different methods by visualizing the Centered Kernel Alignment (CKA) similarity matrix of each method in Appendix E3. We can see that the features learned by DAM are more distinct from one another than other baselines, suggesting that pruning the neurons any further would lead to drop in accuracy. We also observe that the size of the pruned features (with non-zero values) extracted by DAM is quite smaller than what we obtain from Net-Slim and ChipNet using their official implementations. This is because some proportion of pruned channels become nonzero after fine-tuning in Net-Slim and ChipNet. Their current implementations thus result in smaller actual pruning ratios after fine-tuning for practical use than what is reported (before fine-tuning).
>
> ***Comment 4***
>
> Fixing a list of typos and grammatical mistakes.
>
> ***Response***
>
> Thanks very much for finding them. We will fix it in the revised paper.

---

### Official Review · Reviewer_pwT3 · 2021-07-12

**Rating:** 8
**Confidence:** 4

**Summary:**

This paper presents a simple and novel in-training model pruning algorithm to learn sparse feature representation with reduced computational load w/o compromising accuracy. To do this, authors introduce a single learnable parameter to simplify the optimization involved in enforcing L0 sparsity. Authors further provide theoretic analysis on how reducing the single learnable parameter would enforce L0 sparsity. Finally, authors demonstrate the remarkable performance over various applications comparing with the established strong baselines.

**Main Review:**

While authors have theoretically and empirically justified the using a single learnable parameter to achieve L0 sparsity, they only empirically demonstrate the attractive property of permutation invariance via sensitivity analysis.  In addition to the claim “all neurons are symmetrically invariant to each other at initialization and any neuron has an equally good chance of refining themselves to capture useful features later in the training process as any other neuron”, can authors provide analysis why a trivial choice of \mu_{ij} in line 187 would work? For empirical analysis reported in Table 2, what are the corresponding values of the learnable parameters? Any guidance on the “optimal” sparsity for a given task and data set that would balance the trade-off between accuracy and network size?

In addition to reducing resource consumption, network pruning is also to improve generalization to new examples, either In-Distribution (ID) or Out-Of-Distribution (OOD), it seems that all the evaluations are based on ID validation sets in a relatively smaller scale (up to CIFAR-100), additional experiments using new larger data sets (e.g., mini-ImageNet or ImageNet) would further demonstrate the advantages of your network pruning approach.


**Time Spent Reviewing:**

4 hours

---

> ### Author Response · Authors · 2021-08-11
> **Author Response to Reviewer pwT3**
>
> We thank the reviewer for their valuable comments. Here are our point-wise responses to the four comments of the reviewer.
>
> ***Reviewer's Comment 1:***
>
> While authors have theoretically and empirically justified the use of a single learnable parameter to achieve $L_0$ sparsity, they only empirically demonstrate the attractive property of permutation invariance via sensitivity analysis. In addition to the claim “all neurons are symmetrically invariant to each other at initialization and any neuron has an equally good chance of refining themselves to capture useful features later in the training process as any other neuron”, can authors provide analysis why a trivial choice of $\mu_{ij}$ in line 187 would work?
>
> ***Response:***
>
> Thank you for this comment. To theoretically understand why randomly permuting the neuron orders, $\mu_{ij}$, leads to near-equivalent empirical performance in Table 2, let us look at an attractive property common to most of the widely used methods for initializing neural network weights, which is their i.i.d. nature. In particular, the weights at every neuron are independent and identically distributed by commonly used initialization methods when training from scratch. As a result of this intrinsic symmetry of neuron weights, all neurons are invariant to each other at the time of initialization, i..e, any permutation of neurons would result in a network setting that is equally likely to be generated by the initialization procedure as the original setting. Since we cannot discriminate between the importance of one neuron over another at the time of initialization without looking at the labels, we use a trivial choice of neuron order $\mu_{ij}$ that is linearly proportional to the position (or index) j of the neuron in the layer, where the proportionality constant is given by $k/n_i$. For example, if we have $n_i = 5$ neurons in a layer and $k = 1$, then neuron 1 will receive $\mu_1 = 0.2$, neuron 2 will receive $\mu_2 = 0.4$, and so on. We will include this discussion in the revised motivation of our work.
>
> ***Reviewer's Comment 2:***
>
> For empirical analysis reported in Table 2, what are the corresponding values of the learnable parameters?
>
> ***Response:***
>
> In Figure 9 of the Supplementary Materials, we provide visualizations of the pruned architectures for Pre-Resnet-164 for 5 runs with different random neuron orderings, where we can see that the resulting architectures are similar to each other. This along with the results in Table 2 of the main paper indicate that the training of the learnable parameters is invariant to permutation of neuron ordering.
>
> ***Reviewer's Comment 3:***
>
> Any guidance on the “optimal” sparsity for a given task and data set that would balance the trade-off between accuracy and network size?
>
> ***Response:***
>
>  This is a very important yet hard question. While we ideally want to infer the “optimal” sparsity automatically from data, this is currently not possible using existing methods for learning compact representations including our proposed work, as the optimal sparsity in a problem depends on several factors including the choice of backbone network, the nature of problem (structured pruning vs. representation learning), and the type of dataset. One simple idea to empirically guide the selection of optimal sparsity in our DAM formulation is to plot the validation accuracy vs. network sparsity by repeating the training process of DAM with different configurations of the hyperparameter lambda. The knee of this curve would represent the “optimal” sparsity for that specific problem setup.
>
> ***Reviewer's Comment 4:***
>
> In addition to reducing resource consumption, network pruning is also to improve generalization to new examples, either In-Distribution (ID) or Out-Of-Distribution (OOD), it seems that all the evaluations are based on ID validation sets in a relatively smaller scale (up to CIFAR-100), additional experiments using new larger data sets (e.g., mini-ImageNet or ImageNet) would further demonstrate the advantages of your network pruning approach.
>
> ***Response:***
>
> We thank the reviewer for this comment. We are using the same evaluation setup including choice of benchmark datasets as that used in previous works on structured network pruning (e.g., ChipNet). However, in response to reviewer comments, we were able to generate some new results for the Tiny-ImageNet (http://cs231n.stanford.edu/reports/2017/pdfs/930.pdf). For this dataset, we used PreResNet-164 as the backbone network with standard configurations of DAM as presented in Section 5 of the main paper (with additional details in Appendix D.2). We use the AdaBelief optimizer for faster convergence on this larger dataset. The configurations for the optimizer are:
>
> ```python
> AdaBelief(
> model.parameters(),
> lr=1e-3,
> eps=1e-16,
> betas=(0.9,0.999),
> weight_decay=1e-3,
> weight_decouple = True,
> rectify = False
> )
> ```
> We trained the models from scratch for 200 epochs and obtained the following results:
>
> |  Lambda  | Top-1 Test Accuracy |Channels pruned (%) |
> |:--:|:--:|:--:|
> |Unpruned |52.72 | 0.00 |
> | 0.1 | 52.75   | 6.03 |
> | 0.3 | 52.51 | 20.05 |
> | 0.5 | 52.99 | 30.91 |
> | 0.7 | 52.11 | 38.71 |
> | 1.0 | 52.89 | 45.94 |
> | 2.0 | 53.17 | 68.52 |
>
>
> We can see that DAM is able to consistently maintain the test accuracy on TinyImageNet even at extreme levels of network pruning close to 68%. This demonstrates the ability of DAM to achieve useful generalization on larger datasets going beyond CIFAR. Note that these results were generated without any data augmentations, label smoothings, transfer learning, and other advanced procedures employed by state-of-the-art methods developed for TinyImageNet, which explains the lower range of test accuracies obtained in the above table. However, the goal of our experiment was to only demonstrate the ability of DAM to perform pruning along with maintaining test accuracy on larger datasets, instead of achieving state-of-the-art performance with the most advanced architectures. We will include these results in the Supplementary Materials of the paper.

---

### Official Review · Reviewer_69J5 · 2021-07-16

**Rating:** 6
**Confidence:** 3

**Summary:**

This paper proposes a gradual structured pruning method, DAM, that can achieve good performance with various applications. The paper conducts extensive experiments to support the effectiveness of the proposed methods. The method part is reasonable and makes sense.

**Limitations And Societal Impact:**

No. I agree with the authors that the limitations can be good future work directions.

**Main Review:**

I generally like the simplicity of the proposed method and the empirical results are promising.

What I don't like is the method is proposed heuristically and suddenly. I want to read more reasoning parts of the proposed method. For instance, is this method stemmed from other disciplines or just heuristically? Can we use sigmoid function rather than tanh in equation 2?

Moreover, I encourage the authors to add more unstructured sparsity methods in the related work. First,  gradual magnitude pruning [1,2] also gradually prunes the model to the target sparsity during training, which should be added to the related work. An important branch of unstructured sparsity methods, dynamic sparse training [3,4,5], is ignored. Even though it is an unstructured method, the sparse connectivity is allowed to be gradually optimized during training.

[1] Zhu, Michael, and Suyog Gupta. "To prune, or not to prune: exploring the efficacy of pruning for model compression." arXiv preprint arXiv:1710.01878 (2017).
[2]Gale, Trevor, Erich Elsen, and Sara Hooker. "The state of sparsity in deep neural networks." arXiv preprint arXiv:1902.09574 (2019).
[3] Mocanu, Decebal Constantin, et al. "Scalable training of artificial neural networks with adaptive sparse connectivity inspired by network science." Nature communications 9.1 (2018): 1-12.
[4] Evci, Utku, et al. "Rigging the lottery: Making all tickets winners." International Conference on Machine Learning. PMLR, 2020.
[5] Liu, Shiwei, et al. "Do we actually need dense over-parameterization? in-time over-parameterization in sparse training." arXiv preprint arXiv:2102.02887 (2021).

**Time Spent Reviewing:**

5 hours

---

> ### Author Response · Authors · 2021-08-11
> **Author Response to Reviewer 69J5**
>
> We thank the reviewer for their insightful comments. Here are our point-wise responses to the three comments of the reviewer.
>
> ***Reviewer’s Comment 1:***
>
>  I want to read more reasoning parts of the proposed method. For instance, is this method stemmed from other disciplines or just heuristically?
>
> ***Response:***
>
>  Thank you for this comment. The primary motivation behind DAM is to drastically simplify the process of structured pruning by only allowing a single learnable parameter to control the pruning of all neurons in a layer, rather than allowing every neuron to be pruned differently using a distinct learnable parameter, as is the case in current state-of-the-art methods for structured pruning. This simplification is possible only because we are able to order the neurons in every layer during initialization, such that neurons with lower order numbers are selectively (or discriminatively) pruned, while neurons with higher order numbers are selectively retained and refined during the training process. While finding the optimal ordering of neurons to perform selective pruning is a difficult problem for any general setting of neuron weights, we leverage the fact that most of the commonly used methods for initializing neural network weights employ i.i.d procedures, i.e., the weights at every neuron are independent and identically distributed when training from scratch. As a result of this intrinsic symmetry of neuron weights, all neurons are invariant to each other at the time of initialization and hence any random ordering of neurons will be equally good to perform selective pruning (we empirically demonstrate this in Section 5 with additional details in Appendix E4). This is at the basis of our choice of a very simple scheme for neuron ordering in DAM using the index of neurons at every layer as the neuron order. We hypothesize the simplicity of our pruning approach (by using a shared learnable parameter to control the puning of all neurons in a layer) as one of the primary reasons behind the superior performance of DAM, further motivating future investigations to reduce the complexity of the process of structured pruning, which is likely more complex than necessary in existing formulations. While our work is completely novel and is not directly inspired by any of the existing methods for structured pruning, there are several interesting connections between DAM and other related concepts in this area, as described in Section 6. For example, our approach shares a similar intuition as Dropout, where randomly dropping neurons during training forces other neurons to “re-distribute” themselves and pick up the features of the dropped neurons. This is similar to the “re-adaptation” of weights at the active neurons (with high neuron order numbers) in DAM training, while the mask values of neurons in the transitioning zone are gradually dropped by the shifting of the gate function. We will revise the motivation of our work to include this discussion.
>
> ***Reviewer’s Comment 2:***
>
>  Can we use sigmoid function rather than tanh in equation 2?
>
> ***Response:***
>
>  Using a sigmoid function rather than tanh in the gate function is a very interesting suggestion. In fact, our initial design of the gate function was based on sigmoid. However, we found that sigmoid cannot not be used as a valid choice for the gate function as it does not have the “hard-thresholding” property, i.e., the gate function value of sigmoid does not remain exactly 0 before a certain threshold value of input is reached. This property is important to enforce strict L0 sparsity, otherwise the neurons would still remain active with small non-zero gate values across a large range of inputs. In general, there are three properties that we desire in an ideal gate function: (a) it should be exactly 0 before reaching a certain threshold value of input, (b) it should be monotonically increasing, and (c) it should have a parameter to control the steepness of the gate function. While we found RELU-tanh to exhibit stable training dynamics across different datasets in our experiments, other choices of gate functions can also be explored (e.g., hard-sigmoid). We will include this discussion about the choice of gate function in the Supplementary Materials. Additional theoretical discussions on the steepness of the gate functions and its effect on training dynamics can be found in Appendix A3.
>
> ***Reviewer’s Comment 3:***
>
>  Moreover, I encourage the authors to add more unstructured sparsity methods in the related work. First, gradual magnitude pruning [1,2] also gradually prunes the model to the target sparsity during training, which should be added to the related work. An important branch of unstructured sparsity methods, dynamic sparse training [3,4,5], is ignored. Even though it is an unstructured method, the sparse connectivity is allowed to be gradually optimized during training.
>
> [1] Zhu, Michael, and Suyog Gupta. "To prune, or not to prune: exploring the efficacy of pruning for model compression." arXiv preprint arXiv:1710.01878 (2017). [2]Gale, Trevor, Erich Elsen, and Sara Hooker. "The state of sparsity in deep neural networks." arXiv preprint arXiv:1902.09574 (2019). [3] Mocanu, Decebal Constantin, et al. "Scalable training of artificial neural networks with adaptive sparse connectivity inspired by network science." Nature communications 9.1 (2018): 1-12. [4] Evci, Utku, et al. "Rigging the lottery: Making all tickets winners." International Conference on Machine Learning. PMLR, 2020. [5] Liu, Shiwei, et al. "Do we actually need dense over-parameterization? in-time over-parameterization in sparse training." arXiv preprint arXiv:2102.02887 (2021).
>
> ***Response:***
>
>  Thank you very much for these suggestions. We feel that covering these papers on unstructured network pruning would enrich our related works section and we will revise Section 2 to include them.

---

### Official Review · Reviewer_5AFY · 2021-07-17

**Rating:** 6
**Confidence:** 4

**Summary:**

The authors propose a single stage pruning method (DAM) that jointly prunes and refines weights during training. The method uses a monotonically increasing gate function for the neurons/channels in each layer with one trainable parameter. The gate function only discriminates neurons based on the position of them in the layer. The proposed method achieves reasonable results without any fine-tuning.

**Limitations And Societal Impact:**

Yes

**Main Review:**

The proposed method is rather simple, in a positive way, and tries to address an important problem. It does not need fine-tuning which can result in lower training time. The performance of it is evaluated on several applications, like graph representation learning and structured network pruning for image classification, where it shows reasonable performance compared with the baselines. In addition, studies on sensitivity to hyperparameters are provided which are helpful, and detailed experimental setups are included that is a plus.

In the related work, [24] can be applied as a structured pruning through group sparsity, it also considers the L0 problem and can be single stage and versatile. So, the discussion there can be adjusted as the proposed method is not the first single-stage pruning method considering the L0 problem. [24] can also be used as a baseline.

In the contributions, it is stated that the proposed method can be applied to both training from scratch and pre-trained setups. Currently, no experiment is done to support the pre-trained setup. It appears that the success of the method is claimed to be based on neurons being symmetrically invariant to each other at initialization. For a pre-trained network, that symmetry may not hold so it is unclear whether the method would work in that case.

In the recommendation system (where the results are very close) and graph representation examples, it will be helpful to tune the dropout rate or replace the layer with a learnable version like concrete dropout to have a better baseline for comparison to support the paper’s claims.

---After author response---
I thank the authors for their responses. I still think that for the mentioned experiments having a more comparable and better baseline would be helpful. After reading other reviews and responses, I remain positive about the paper.



**Time Spent Reviewing:**

2.5

---

> ### Author Response · Authors · 2021-08-11
> **Author Response to Reviewer 5AFY**
>
> We thank the reviewer for their valuable feedback. Here are our point-wise responses to the three comments of the reviewer.
>
> ***Reviewer's Comment 1:***
>
> In the related work, [24] can be applied as a structured pruning through group sparsity, it also considers the $L_0$ problem and can be single stage and versatile. So, the discussion there can be adjusted as the proposed method is not the first single-stage pruning method considering the L0 problem. [24] can also be used as a baseline.
>
> ***Response:***
>
> Thank you very much for pointing this out. We agree that the method proposed in [24] could be potentially used as a single-stage structured pruning method through group sparsity. We will adjust our discussion to include [24] in the set of related works on structured network pruning in Section 2. However, note that while the group sparsity approach was presented as a potential option in [24], the empirical analysis of the paper only focused on unstructured pruning results. Further, we could not find the group sparsity version on the official implementation of the paper (https://github.com/AMLab-Amsterdam/L0_regularization). Given the time constraints in implementing this version from scratch and the challenges in tuning the hyper-parameters of any implementation (which may not be optimally chosen and hence difficult to be claimed as an accurate representation of the original work), we did not consider the group sparsity version of [24] as a baseline in our paper. However, note that a very recent state-of-the-art on structured network pruning, ChipNet, which appeared in ICLR 2021 two weeks before the NeurIPS deadline, performed a comparison with [24] and showed that ChipNet significantly outperforms this baseline. Since we compare our results with ChipNet, we hope this serves as an indirect line of comparison between DAM and [24].
>
> ***Reviewer's Comment 2:***
>
> In the contributions, it is stated that the proposed method can be applied to both training from scratch and pre-trained setups. Currently, no experiment is done to support the pre-trained setup. It appears that the success of the method is claimed to be based on neurons being symmetrically invariant to each other at initialization. For a pre-trained network, that symmetry may not hold so it is unclear whether the method would work in that case.
>
> ***Response:***
>
>  We thank the reviewer for this very careful observation. Indeed, DAM is not compatible with pre-trained networks because of the potential violation of neuron symmetry in such networks, which is fundamental to our current choice of random neuron ordering in DAM. We will revise the paper to correct our statement and clarify that the current formulation of DAM is only compatible with the “training from scratch” regime. Further, we also found this comment on compatibility with pre-training very helpful for postulating novel extensions of our basic DAM formulation that can handle pre-training. In particular, instead of using the simple approach of random neuron ordering, we can formulate advanced ways of ordering neurons in DAM that do not rely on neuron symmetry during initialization, e.g., based on the magnitudes of weights of every neuron or the value of mutual information between neuron activations and outputs. We thus anticipate the basic DAM formulation proposed in our paper to open novel avenues of research in structured network pruning by finding effective neuron orderings for different initializations of network weights (including pre-trained weights). We will include this discussion as a future direction of work in Section 6.
>
> ***Reviewer's Comment 3:***
>
> In the recommendation system (where the results are very close) and graph representation examples, it will be helpful to tune the dropout rate or replace the layer with a learnable version like concrete dropout to have a better baseline for comparison to support the paper’s claims.
>
> ***Response:***
>
>  We agree with the reviewer that improving the baseline models used as backbone networks in DAM can further improve our paper’s results on the recommendation system and graph representation learning experiments. However, note that our primary goal in both these experiments was to show that DAM can improve the performance of any backbone network and achieve state-of-the-art (SOTA) performance, without making any refinements in the backbone network. Specifically, for recommendation systems, our focus was to show that DAM, when applied on a SOTA method used as backbone network, can lead to better parameter efficiency without any loss of accuracy (see Table 2). On the other hand, for graph representation learning, our focus was to show that even when DAM is applied on a very simple backbone architecture (GAE), it can achieve SOTA performance on benchmark datasets and outperform highly sophisticated baseline methods developed for this problem by a significant margin for Cora, CiteSeer, and PubMed datasets. While future work can focus on using more advanced variants of backbone networks in both these problems to possibly achieve even higher SOTA performance, our interest in this work was to only demonstrate the power of DAM in learning more efficient and generalizable feature representations  using any choice of backbone network.

---

### Official Review · Reviewer_VSsj · 2021-07-18

**Rating:** 7
**Confidence:** 4

**Summary:**

This paper proposes a novel structured pruning method called DiscriminAtive Masking (DAM) for neuron-level sparsity. A relu-tanh gating function is proposed to apply on the neuron activations, and gradually zeroing out the neurons at lower order by introducing a L0 norm. The proposed method is a single stage method, and does not require further after-prune finetuning. Extensive experiments demonstrate the effectiveness of the proposed method

**Limitations And Societal Impact:**

- It is understandable that lambda controls the compactness of tthe representations or the pruning sparsity. But it seems that what is the exact relationship between lambda and the resulting compactness is not clear and could depend on the specific dataset. For example, if we want to achieve 20% sparsity, in one dataset lambda needs to be 10.512, but in another dataset it could be 5.123, etc. Is it a right understanding? In practise, how should be find the right lambda if we have an sparsity in mind?

**Main Review:**

The proposed DAM method for learning compact representations is novel. Unlike previous method in applying L1 norm as sparsity-inducing regularization followed by further pruning and finetuning, the proposed method applies the L0 norm and does not need additional step of finetuning with the proposed gating function and the neuron ordering. And it does not require combinatorial search, and can be trained with simple gradient descent.

Extensive experiments have been conducted to demonstrate the effectiveness of the proposed method. Specifically, the authors conduct experiments for representation learning problems, including dimensionality reduction, recommendation, graph representation, etc., in addition to the structure network pruning. The results show that even better performance can be achieved with more compact representations with the proposed DAM for recommendation problem. For structure pruning, it is shown to perform better than the state-of-the-art ChipNet. Stability analysis and permutation invariance experiments have also been conducted.

Overall, I think the paper is of high quality, and contain significant novelty.

**Time Spent Reviewing:**

2

---

> ### Author Response · Authors · 2021-08-11
> **Author Response to Reviewer VSsj**
>
>
> ***Reviewer's Comment:***
>
> It is understandable that lambda controls the compactness of the representations or the pruning sparsity. But it seems that what is the exact relationship between lambda and the resulting compactness is not clear and could depend on the specific dataset. For example, if we want to achieve 20% sparsity, in one dataset lambda needs to be 10.512, but in another dataset it could be 5.123, etc. Is it a right understanding? In practise, how should be find the right lambda if we have an sparsity in mind?
>
> ***Response:***
>
> We thank the reviewer for pointing this out and we fully agree with the reviewer that to achieve a specific sparsity in mind, our current formulation requires running DAM with different values of $\lambda$, which are directly proportional to the resulting compactness but are dependent on the choice of dataset. However, in contrast to $\lambda$, there is a direct correspondence between the value of the learnable gate offset parameter $\beta_i$ (that is minimized at every epoch) and the resulting $L_0$ sparsity (see Equation 3 in the paper). Hence, a simple extension of our current DAM formulation to make it budget-aware, i.e., to stop the training process once we arrive at a desired level of sparsity, is to start from a large value of lambda (to provide sufficient scope for aggressive pruning) and keep monitoring the level of sparsity at every layer during the training process. As soon as the target sparsity mark is achieved, we can freeze $\beta_i$ at every layer thus essentially halting the pruning process (note that the network architecture remains immutable if $\beta_i$ is not further lowered). We could not complete all the results of this budget-aware extension of DAM due to time constraints but we definitely consider this a valuable future direction of work, as mentioned in Section 6. We will include this complete description in Section 6 of our revised paper. We also want to point out that our results are quite stable to the choice of $\lambda$ (as shown in Appendix C) and we show that our pruning keeps continuing even with aggressively large values of $\lambda$ (see Section 4.4 of main paper and Appendix E1), which further demonstrates that DAM does not suffer from the saturation effects of $L_1$ based regularization methods.

---

### Author Response · Authors · 2021-08-11
**Response to All Reviewers**

We sincerely thank all the reviewers for their feedback. We are encouraged that the reviewers found our work:
- significantly novel,
- simple to understand,
- provides theoretical connections between our learning objective and minimizing L0 sparsity,
- includes extensive evaluation with strong baselines over many problems and datasets with sensitivity analysis, and
- provides joint analysis of representation learning and structured pruning problems.

In response to the reviewer’s comments, here is a summary of the revisions we will be making to our paper:
- We will expand our discussion on the budget-aware variant of DAM as a future direction of research in Section 6.
- We will include [24] in the set of related works on single-stage structured network pruning in Section 2.
- We will adjust our discussion of DAM by clarifying that our current formulation is only compatible with “training from scratch” regime. We will also include a discussion on possible future extensions of our basic DAM formulation to work with pre-trained networks in Section 6.
- We will revise the motivation of our work to include more details on why our approach is invariant to permutations of neuron orderings.
- We will add more information regarding the choice of alternate gate functions in the Supplementary Materials.
- We will enrich our related works section on unstructured pruning by including the papers pointed out by the reviewers.
- We will revise our abstract to further clarify that the focus of our work is on performing a joint analysis of the problems of representation learning and structured pruning from the unified perspective of learning compact representations.
- We will add our new results on TinyImagenet in the Supplementary Materials to demonstrate the performance of DAM pruning on larger datasets going beyond CIFAR.
- We will fix all the typos pointed out by reviewers in our revised manuscript.

Our detailed response to each of the reviewer’s comments can be found in our individual responses below.

---

### Decision · Program_Chairs · 2021-09-27

**Decision:**

Accept (Poster)

**Comment:**

The submission proposes a method for learning a network with sparse parameters.  The key idea is to optimize an L0 penalized objective (1).  This is non-differentiable, so instead a ReLU is applied to a parametrized tanh scaling (2) which will selectively zero out some fraction of the weights, where the fraction is monotonic in the beta parameter.  This leads to the main result of the paper (3), which explicitly demonstrates this relationship.  I was initially confused by the comment of Reviewer VSsj who stated that an L0 constraint (non-differentiable) could be trained by gradient descent.  One clue is that the "equality" in (3) includes a ceiling function, implicitly showing that there is indeed a relaxation of the L0 constraint, through the tanh function, albeit a different relaxation than is given by e.g. the L1 relaxation.

Overall, a majority of the reviewers feel that this paper is ready to be accepted at NeurIPS.  It is an interesting and clearly presented contribution to learned network sparsity.  Comparison to baseline methods is on the weak side, but overall this paper is acceptable at NeurIPS.